# *Helicobacter pylori* diversification during chronic infection within a single host generates sub-populations with distinct phenotypes

Laura K. Jackson[1,2], Barney Potter[3], Sean Schneider[2], Matthew Fitzgibbon[4], Kris Blair[1,2], Hajirah Farah[2,5], Uma Krishna[6], Trevor Bedford[2,3], Richard M. Peek Jr[6], Nina R. Salama[1,2,5]*

1 Molecular and Cellular Biology Graduate Program, University of Washington, Seattle, WA, United States of America, 2 Human Biology Division, Fred Hutchinson Cancer Research Center, Seattle, WA, United States of America, 3 Vaccine and Infectious Disease Division, Fred Hutchinson Cancer Research Center, Seattle, WA, United States of America, 4 Genomics & Bioinformatics Shared Resource, Fred Hutchinson Cancer Research Center, Seattle, WA, United States of America, 5 Department of Microbiology, University of Washington School of Medicine, Seattle, WA, United States of America, 6 Division of Gastroenterology, Department of Medicine, Vanderbilt University Medical Center, Nashville, TN, United States of America

* nsalama@fredhutch.org

**Data Availability Statement:** All data is in the manuscript and the supporting information files. Sequence reads are available on NCBI SRA

## Abstract

*Helicobacter pylori* chronically infects the stomach of approximately half of the world's population. Manifestation of clinical diseases associated with *H. pylori* infection, including cancer, is driven by strain properties and host responses; and as chronic infection persists, both are subject to change. Previous studies have documented frequent and extensive within-host bacterial genetic variation. To define how within-host diversity contributes to phenotypes related to *H. pylori* pathogenesis, this project leverages a collection of 39 clinical isolates acquired prospectively from a single subject at two time points and from multiple gastric sites. During the six years separating collection of these isolates, this individual, initially harboring a duodenal ulcer, progressed to gastric atrophy and concomitant loss of acid secretion. Whole genome sequence analysis identified 1,767 unique single nucleotide polymorphisms (SNPs) across isolates and a nucleotide substitution rate of $1.3 \times 10^{-4}$ substitutions/site/year. Gene ontology analysis identified cell envelope genes among the genes with excess accumulation of nonsynonymous SNPs (nSNPs). A maximum likelihood tree based on genetic similarity clusters isolates from each time point separately. Within time points, there is segregation of subgroups with phenotypic differences in bacterial morphology, ability to induce inflammatory cytokines, and mouse colonization. Higher inflammatory cytokine induction in recent isolates maps to shared polymorphisms in the Cag PAI protein, CagY, while rod morphology in a subgroup of recent isolates mapped to eight mutations in three distinct helical cell shape determining (*csd*) genes. The presence of subgroups with unique genetic and phenotypic properties suggest complex selective forces and multiple niches within the stomach during chronic infection.

database (BioProject accession: PRJNA633860, <https://www.ncbi.nlm.nih.gov/sra/PRJNA622860>. All datasets, config files, documentation, and scripts used in this analysis or to generate figures are available publicly on our Github page <https://github.com/salama-lab/Hp-J99>. The interactive tree can be found at the Nextstrain community site <https://nextstrain.org/community/salama-lab/Hp-J99.

**Funding:** This research was supported by the National institute of Allergy and Infectious Diseases <https://www.niaid.nih.gov> (R01 AI054423, NRS), the National Cancer Institute <https://www.cancer.gov> (T32 CA009657, LKJ), the Institute of Diabetes and Digestive Kidney Diseases <https://www.niddk.nih.gov> (P30 DK056456-16S1), The National Institute of General Medical Sciences <https://www.nigms.nih.gov> (R35 GM119774, TB), and the Genomic & Bioinformatics and Comparative Medicine Shared Resources of the Fred Hutch/University of Washington Cancer Consortium funded by the National Cancer Institute <https://www.cancer.gov> (P30 CA015794). TB is a Pew Biomedical Scholar. The funders had no role in study design, data collection and analysis, decision to publish, or preparation of the manuscript.

**Competing interests:** The authors have declared that no competing interests exist.

## Author summary

*Helicobacter pylori*, one of the most common bacterial pathogens colonizing humans, is the main agent responsible for stomach ulcers and cancer. Certain strain types are associated with increased risk of disease, however many factors contributing to disease outcome remain unknown. Prior work has documented genetic diversity among bacterial populations within single individuals, but the impact of this diversity for continued bacterial infection or disease progression remains understudied. In our analysis we examined both genetic and functional features of many stomach isolates from a single individual infected over six years. During these six years the subject shifted from having excess acid production and a duodenal ulcer to lower acid production from gastric atrophy. The 39 isolates form sub-populations based on gene sequence changes that accumulated in the different isolates. In addition to having distinguishing genetic features, these sub-populations also have differences in several bacterial properties, including cell shape, ability to activate immune responses, and colonization in a mouse model of infection. This apparent functional specialization suggests that the bacterial sub-populations may have adapted to distinct niches within the stomach during chronic infection.

## Introduction

*Helicobacter pylori* is a bacterial pathogen that colonizes the human gastric mucosa of approximately half of the world's population [1]. Infections persist throughout life without intervention and can lead to gastric and duodenal ulcers, MALT lymphoma, and gastric cancer in a subset of individuals [2, 3]. *H. pylori* exhibits marked genetic diversity compared to other bacterial pathogens, which can impact treatment efficacy and disease severity [4–7]. Typically, antibiotics and proton pump inhibitors are employed for treatment, but variable prevalence of antibiotic resistance across populations make implementing a single treatment regimen difficult [8, 9]. Strain-specific genotypes also contribute to increased disease risk within distinct ethno-geographic populations. Individuals with strains carrying the Cag pathogenicity island (Cag PAI), encoding a type IV secretion system (T4SS) and effector toxin CagA, have an increased risk of gastric cancer [10, 11]. Cag PAI encoded genes, *cagA* and *cagY*, exhibit significant allelic variation between individuals and have been identified as targets of positive selection within the global population [12]. Both CagA and CagY have been shown to modulate the host inflammatory response [13, 14]. Recombination events within *cagY*, which encodes a structural component of the Cag T4SS with homology to the VirB10 component of other T4S systems, modifies secretion of inflammatory cytokines from epithelial cells [15, 16]. CagA alters host responses through its interaction with intracellular kinases leading to the activation of the NFκB pathway [17, 18]. In addition to Cag PAI genes, certain alleles of vacuolating cytotoxin, *vacA*, and frequent phase variation as well as recombination mediated gain and loss of outer membrane protein (OMP) adhesins BabA, SabA, and HopQ, have been linked to strain differences in pathogenesis [19–22].

Several mechanisms promote genomic diversification. Although *H. pylori* does encode several transcriptional regulators, much of gene regulation occurs through genomic alterations [23]. *H. pylori* has several phase variable genes whose expression is altered due to slipped strand mispairing in homo-polymeric tracts [24, 25]. In addition, *H. pylori* has a somewhat elevated baseline mutation rate compared to other bacteria; $10^6 - 10^8$ substitutions/site/generation compared to the $10^9$ substitutions/site/generation reported for *Escherichia coli* [26–28]. This is

due to absent mismatch repair genes and deficiencies in the exonuclease domain of Pol1 [29]. However, base-excision repair is robust, preventing hypermutator phenotypes [30]. Variation is largely driven by high rates of intra and inter-genomic recombination. Intragenomic recombination can alter protein expression via gene conversion among paralogous families of outer membrane proteins [25]. Additionally, as a naturally competent bacterium, *H. pylori* incorporates DNA from genetically distinct strains into its chromosome, further varying gene content and sequence [31, 32].

The human stomach is the only known niche for *H. pylori;* therefore, the breadth of genomic diversity across global populations likely reflects adaptation to individual host stomach environments [33]. More recently, genetic diversity within a single host has also become appreciated, suggesting the existence of structured niches within the stomach with distinct selective pressures [34–37]. *H. pylori* can colonize the epithelial surface of the inner gastric mucus layer, form cell adherent microcolonies, and penetrate into the gastric glands in both the antrum and corpus (Fig 1) [38, 39]. The antrum and corpus have distinct gland architecture and cell type composition, providing unique challenges to bacterial survival. Gastric environments also change during lifelong infection. While most acute infections start in the antrum where the pH is closer to neutral, *H. pylori* can expand into the corpus [40, 41]. Changes in bacterial localization are associated with histologic changes, including loss of the parietal cells (gastric atrophy), a risk factor for the development of gastric cancer [42]. These changes are accompanied by fluctuations in immune responses and alteration of glycosylation patterns affecting OMP-receptor binding to cell surface and mucus [19, 24, 43].

Whole genome sequencing of sequential isolates from individual infections detected accumulation of point mutations and frequent recombination among co-colonizing isolates. Within host diversity is acquired in a clock-like manner with average estimated mutation rate of $2.5 \times 10^{-5}$ substitutions per site per year. However, in the majority of individuals recombination is the main driver of within-host diversification, which introduces changes at a median rate of $5.5 \times 10^{-5}$ substitutions per initiation site per year [44, 45]. The rate of polymorphisms introduced via recombination compared to mutation (r/m rates) vary widely between individuals and likely depends on the presence of multiple co-infecting strains [46, 47]. Phylogenetic studies of isolates within a single individual have observed signatures of diversifying selection in support of selective pressures during chronic stomach colonization [7, 37, 45]. However,

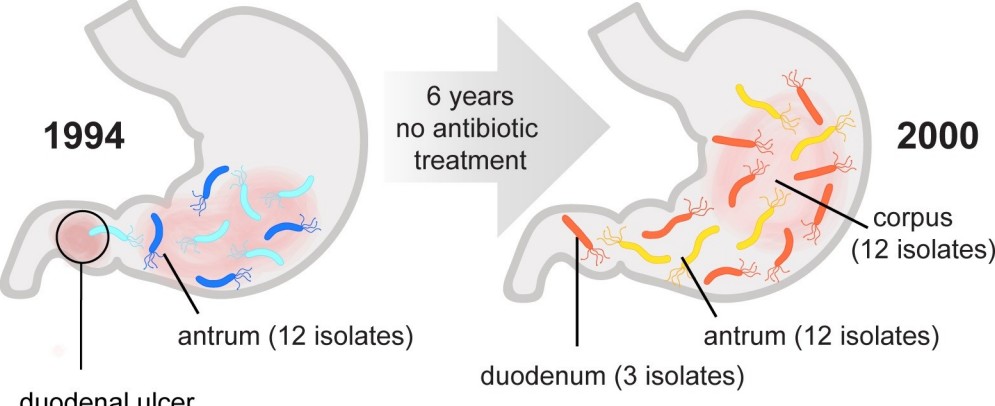

**Fig 1. Sampling utilized to characterize genetic and phenotypic diversification of infecting population over six years.** This study leverages a collection of *H. pylori* isolates obtained from a treatment-naïve subject initially presenting with a duodenal ulcer at two different time points over a 6-year period of infection. A total of 12 isolates were analyzed from a single antral biopsy in 1994, and a total of 27 isolates were analyzed from single corpus, antrum, and duodenum biopsies collected in 2000 as indicated. In 2000 the subject displayed corpus atrophic gastritis and elevated stomach pH.

phenotypic variation in infecting populations over time has not been well studied. To define both genetic changes that occur during infection and their functional consequences, we leveraged *H. pylori* isolates from a single individual collected at two time points spanning 6 years (1994–2000). One of multiple isolates obtained from culture of a single antral biopsy in 1994, J99, has previously been sequenced and a complete reference genome is available [48]. This same individual, who had not been successfully treated for *H. pylori*, underwent a repeat endoscopy performed in 2000 from which single biopsies from the corpus, antrum, and gastric metaplasia in the duodenum were cultured and additional *H. pylori* isolates recovered. A subset of isolates from the second time point (yr 2000) were analyzed by PCR microarray as part of a study highlighting diversity of isolates from a single individual [34].

Here we combined whole genome sequence analysis of multiple isolates from this subject with extensive phenotypic characterization to explore the rates and extent of genetic and phenotypic diversification within a single host. In this individual, during six years of chronic colonization, isolates adapted to occupy at least two distinct niches within the stomach reflected by differential ability to colonize a mouse model. We identified the genetic basis for modulation of Cag-dependent inflammatory cytokine induction and morphologic diversification. Neither of these phenotypes fully account for the differences in colonization of the mouse model, highlighting the multifactorial selection pressures operant during chronic stomach colonization.

## Results

### Whole genome sequencing detects within-host genetic diversification of bacterial populations

For this study we analyzed 39 isolates from two distinct sampling time points. At the time of the original biopsy (yr 1994), the source individual had a duodenal ulcer, indicative of *H. pylori* infection localized to the antrum and consistent with recovery of multiple single colonies from the single antral biopsy processed for culture. Six years later (yr 2000), after refusing antibiotic therapy, additional single colony isolates were collected from distinct biopsy sites. At this time, this individual had corpus predominant gastritis and signs of gastric atrophy, including decreased production of stomach acid, indicating the spread of infection to the main body of the stomach (Fig 1) [34, 49]. Twelve ancestral isolates, including *H. pylori* strain J99, were all collected in 1994 from the antral biopsy. From the second time point (recent, yr 2000), we analyzed 27 isolates from the antrum (n = 12), corpus (n = 12), and duodenum (n = 3).

In order to measure the genetic diversity of *H. pylori* populations both within each time point and between time points, we performed whole genome sequencing using Illumina MiSeq. Short reads were aligned using the published sequence of J99 as the reference (AE001439). Our J99 isolate differed from the reference J99 (AE001439) at 755 polymorphic sites (625 SNPs, 130 indels); of these 553 polymorphisms were common to all isolates in the collection (S1 Table) and were excluded from downstream analysis. All 39 isolates shared 99.99% average nucleotide sequence identity (ANI) to our J99 strain at shared sites mapping to the reference. By comparison, Alm et al. found J99 shares 92% ANI with strain 26695, originating from a distinct individual and geographic region [48]. High ANI among the isolates in the collection is consistent with either a single diversifying strain population or a mixed infection with highly related strains. Unique SNPs, and insertion and deletion (indel) events detected in the collection are reported in Table 1. Excluding polymorphisms common to all isolates, a total of 1,767 SNPs and 485 indels were identified (Table 1, S1 Table). This sequence variation represents changes introduced by both de-novo mutation and recombination. SNPs were distributed proportionally across coding and intergenic regions. By contrast, indels were biased towards intergenic regions (chi-squared, p-value<0.0001). Depletion of indels within coding

**Table 1. Summary of unique SNPs, Indels detected by WGS among all the isolates (n = 39) and in the subset of recent isolates (n = 27, yr 2000).**

|              | Total[a] | Coding[a] | nS[b] | S[b] | Intergenic[a] |
|--------------|----------|-----------|-------|------|---------------|
| Total SNPs   | 1,767    | 1,632     | 704   | 928  | 135           |
| Recent SNPs  | 1,379    | 1,270     | 536   | 734  | 109           |
|              | Total[a] | Coding[a] |       |      | Intergenic[a] |
| Total Indels | 485      | 298       |       |      | 187           |
| Recent Indels| 372      | 231       |       |      | 141           |

[a]Unique events are labeled as either coding or intergenic.

[b]Events within coding regions are further subdivided into nonsynonymous (nS) or synonymous (S) categories.

regions likely reflects purifying selection due to high potential of indels to disrupt gene function by introducing frameshifts. Of the total unique SNPs and indels detected (n = 2,252), 238 were shared between ancestral and recent populations, while 263 and 1,751 were exclusively found within the ancestral and recent populations, respectively. The high number of polymorphisms unique to recent isolates demonstrates the substantial population divergence that occurred in this patient over time.

To assess extent of genetic diversity within the collection, we calculated the average pairwise genetic distance (π) for unique pairwise comparisons of isolates from the same time point (within time point) to isolates from different time points (between time point). For between time point comparisons, only antral isolates (n = 24) were used to reduce potential confounding effects introduced from comparing isolates from different anatomical locations. The average genetic distances (π, nucleotide differences/site) of within time point pairs was $6.75 \times 10^{-5}$, while π of between time point pairs was $8.23 \times 10^{-4}$, indicating within host evolution with an average molecular clock rate of $1.3 \times 10^{-4}$ substitutions/site/year (Fig 2A). Overall, recent antral isolates have increased diversity (π = $9.9 \times 10^{-5}$) compared to the ancestral isolates (π = $3.6 \times 10^{-5}$), demonstrating accumulation of genetic diversity during chronic infection (Fig 2B).

## Identification of genomic regions enriched for within-host genetic variation

To identify regions of the genome that accumulate within host variation, we examined enrichment of nonsynonymous SNPs (nSNPs) in specific genes and functional classes assigned by the microbial genome database (MGDB) [50]. Out of the 1,495 genes in the reference sequence, 931(62.2%) are annotated with a functional class (Fig 3A). Enrichment or depletion was determined by comparing the distribution of nSNPs among MGDB classes to expected values based on a normal distribution (Fig 3 and S2 Table). We observed cell envelope genes, including OMPs, accumulated a disproportionate number of nSNPs in both the total dataset of unique nSNPs and the subset of nSNPs unique to recent group of isolates (Fig 3B and 3C and S2 Table). This was expected since other studies that have looked at within host populations have shown OMPs accumulate diversity due to frequent inter and intragenomic recombination [7, 22, 44]. Accordingly, cell envelope diversification may serve a selective advantage in both the acute phase of infection as a mechanism of adaptation to a specific host and in the chronic phase of infection as a mechanism to persist in changing host environments.

The MGDB does not specifically analyze antibiotic resistance genes. While this subject had no known history of antibiotic treatment, the presence of antibiotic resistance to metronizidole, ampicillin, clarithromycin, was previously tested. Four isolates, three antral and one duodenal, were resistant to clarithromycin due to a mutation in the 23S rRNA gene, but all the other isolates were sensitive to all three [34]. To validate, we queried our sequence data for

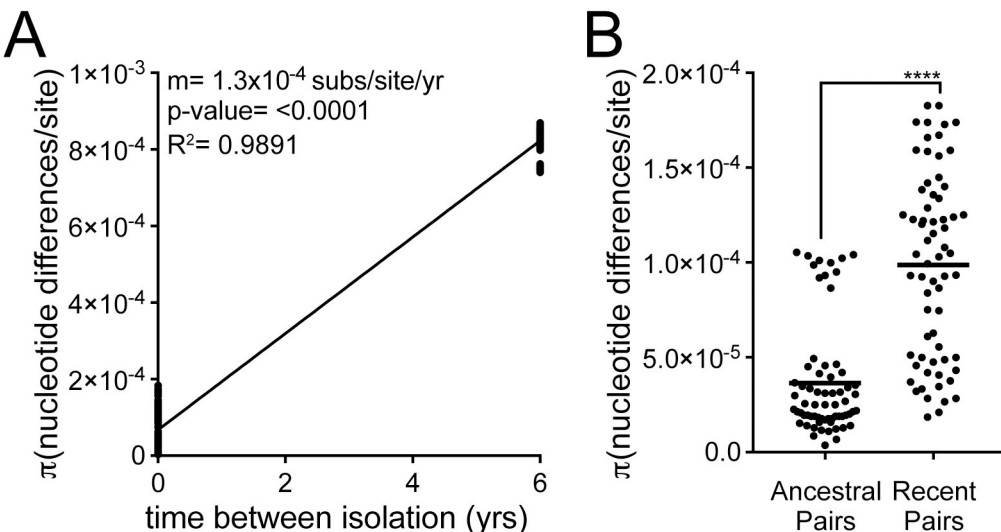

**Fig 2. Pairwise comparisons show within host diversification with increasing diversity of infecting populations over time.** (**A**) Plot shows genetic distance for all pairwise comparisons of antral samples isolated within the same time point (n = 132, time between isolation = 0 yrs) and samples isolated from different time points (n = 144, time between isolation = 6 yrs). Each point is a unique pairwise comparison (n = 276). Linear regression with the slope (m) as the estimation of the molecular clock rate, p-value derived from F-test, and correlation coefficient ($R^2$) shown. (**B**) Each point represents a pairwise comparison between antral isolates within the ancestral population (yr 1994, n = 66) or recent population (yr 2000, n = 66). The average values between all pairwise comparisons in the population (π statistic) is shown with a black bar. Significance was determined using a Student's t-test (****, p<0.0001).

polymorphisms known to confer antibiotic resistance listed in the Pathosystems Resource Integration Center (PATRIC) database. Using these methods, no additional polymorphisms indicative of antibiotic resistance were discovered.

Individual genes acquiring the most genetic variation over the six year period were identified. The number of unique nSNPs among the recent isolates detected for each gene were counted and weighted according to the gene length. The genes most highly enriched for unique nSNPs are listed in Table 2 (S3 Table). Many of the genes identified encode OMPs (*babA*, *sabA*, *sabB*, and *hopQ*) that play roles in adhesion and exhibit variation between and within hosts [51, 52]. Accumulation of nSNPs contributed to allelic diversification. For 7 of the top 15 genes listed, multiple unique alleles were detected within the recent population. For 4 of these genes, jhp1300 and jhp1409, and two outer membrane proteins *sabA* and *babA*, allelic variation in the ancestral population was also detected.

In order to differentiate polymorphism introduced by recombination from repeated mutation, we bioinformatically identified importation sites with ClonalFrameML [53]. A total of 228 importation events across 52 genes were detected in total (S4 Table). All of the genes accumulating an excess of nSNPs listed in Table 2 had at least one detectable recombination event, except jhp0929, a gene of unknown function located within a large highly variable TnPZ associated genomic region called the plasticity zone [54]. The average import size (δ) was 136bp and divergence of import (ν) was $3.9 \times 10^{-2}$ subs/site, suggesting imports were from a genetically distinct strain either not sampled or not present at the time of sampling.

## Genomic diversity within the recent population is not driven by stomach region specific adaptation

To display the genetic relatedness of all the isolates in the collection, a maximum likelihood tree was generated with consensus sequences from the SNP calls using the Nextstrain platform

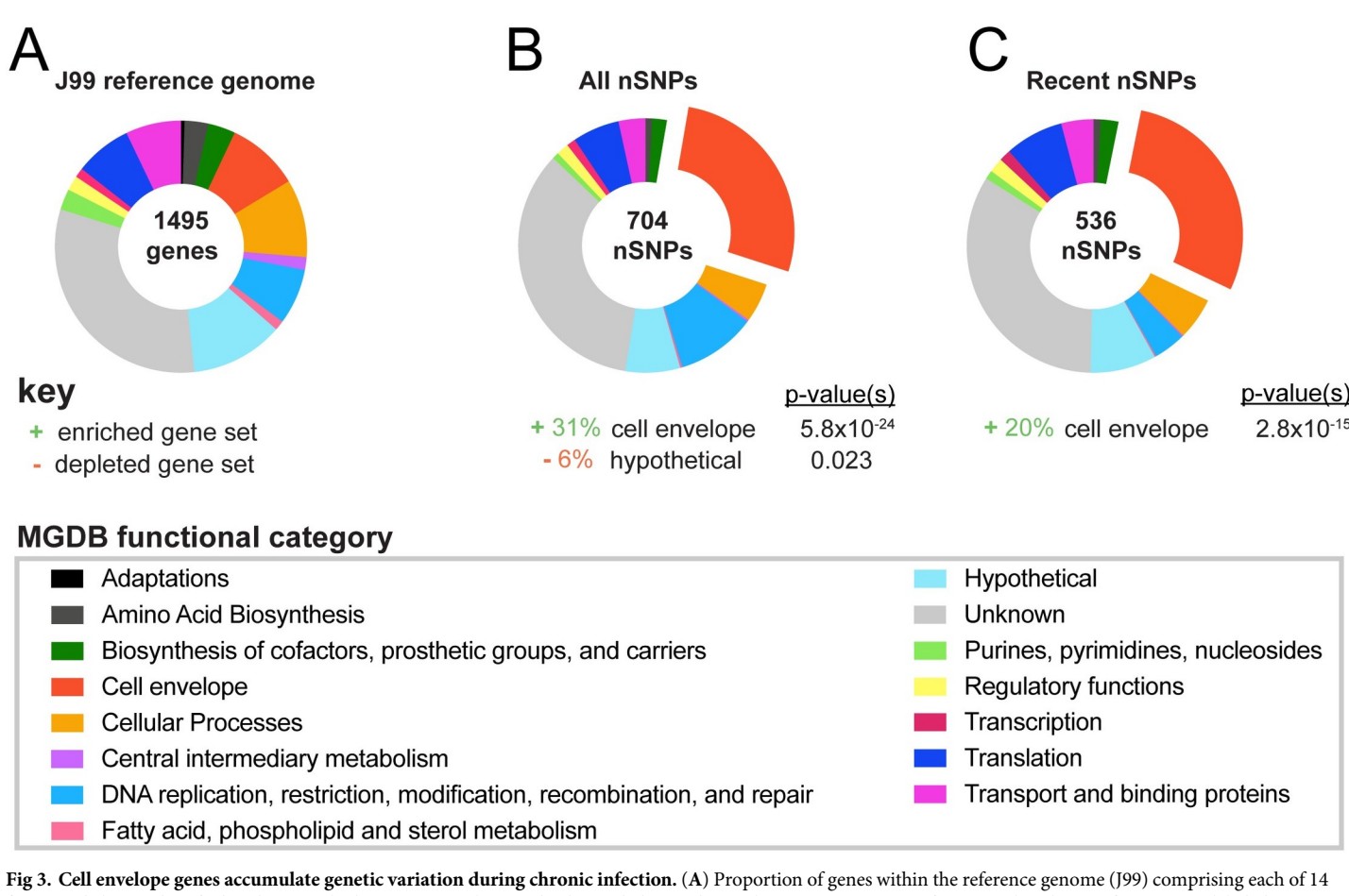

**Fig 3. Cell envelope genes accumulate genetic variation during chronic infection.** (**A**) Proportion of genes within the reference genome (J99) comprising each of 14 functional classes identified in the Microbial Genome Database and color-coded according to key [50]. Percentage of genes with unknown function are labeled in gray. (**B-C**) Proportion of nSNPs from (**B**) the entire dataset (all nSNPs) and (**C**) from the subset unique to recent isolates (recent nSNPs) that fall within each functional class. Categories with statistically significant enrichment or depletion are listed below each chart with associated percentages and p-values ($<0.03$). Fisher's exact tests were used to determine significance and corrected for multiple testing using Benjamini and Hochberg false discovery rate methods.

(Figs 4A and S1A) [55]. Isolates collected from the two separate time points (1994 and 2000) cluster into distinct groups on the tree with a long branch representing an average divergence of $7.8 \times 10^{-4}$ substitutions/site between the two populations. This depiction does not represent a clonal genealogy however, as SNPs included in these analyses can arise from recombination and therefore are not necessarily clonally derived [56].

The inferred molecular clock rate calculated from a time-scaled tree using the root-to-tip distance ($1.35 \times 10^{-4}$ subs/site/year) is similar to the molecular clock rate calculated from pairwise comparisons (Figs 2A, S2A and S2B). Expectedly, this rate is much higher than other within-host estimates that exclude predicted recombination events [44]. To explore the impact of recombination on the tree branch scale and topology, we generated a recombination corrected phylogeny using ClonalFrameML (S1B Fig). We found the branch lengths of the maximum likelihood tree (Figs 4A and S1A) were on average 5 times greater than that of the recombination corrected tree, but general topology remained the same (S1B Fig). The inferred molecular clock rate from the recombination corrected time-scaled tree was $2.2 \times 10^{-5}$ subs/site/year, which is consistent with published estimates (S2D Fig) [44, 47, 53]. We estimated a time to most recent common ancestor (TMRCA) of 1 year for the ancestral population and 1.5 years for the recent population (S2C Fig). Low TMRCA are typically observed in phylogenetic

**Table 2. Genes with excess accumulation of nSNPs during chronic infection.**

| Gene ID[a] | Annotation[a] | MGDB function[a] | Unique nSNPs | Z-scores[b] | Estimated no. of alleles[c] | |
|---|---|---|---|---|---|---|
| | | | | | Ancestral | Recent |
| jhp1300 | | Unknown | 18 | 21.52 | 2 | 19 |
| jhp1103 | *hopQ* | Outer membrane protein | 45 | 14.89 | 1 | 11 |
| jhp1068 | *birA* | Biotin protein ligase | 11 | 10.96 | 1 | 1 |
| jhp0659 | *sabB* | Outer membrane protein | 27 | 8.94 | 1 | 13 |
| jhp0303 | | Translocase subunit | 3 | 7.9 | 1 | 1 |
| jhp1096 | *glnP_1* | Transport and binding proteins | 8 | 7.73 | 1 | 1 |
| jhp1409 | | Unknown | 44 | 7.39 | 2 | 19 |
| jhp0302 | *argS* | Arginine-tRNA ligase | 18 | 6.98 | 1 | 1 |
| jhp0336 | | Unknown | 19 | 5.07 | 1 | 1 |
| jhp1097 | *glnP_2* | ABC transporter permease | 5 | 4.63 | 1 | 1 |
| jhp0634 | | Unknown | 7 | 4.29 | 1 | 1 |
| jhp1102 | | Guanine permease | 9 | 4.26 | 1 | 1 |
| jhp0833 | *babA* | Outer membrane protein | 15 | 4.16 | 2 | 4 |
| jhp0662 | *sabA* | Outer membrane protein | 13 | 4.12 | 4 | 5 |
| jhp0929 | | Unknown | 3 | 4.1 | 1 | 3 |

[a]The top 15 annotated genes with Z-scores > 4 for number of nSNPs uniquely acquired in the yr 2000 group of isolates within single genes are shown.

[b]Z-scores were calculated using number of counts per gene normalized according to gene length.

[c]The estimated number of alleles among 27 isolates in the recent population (1994) and 12 isolates in the ancestral population are shown.

studies of within host evolution of *H. pylori*; and in the absence of antibiotic treatment, this is thought to occur as a result of immune pressure [37, 47].

We defined four distinct subgroups within the collection based on shared genetic characteristics reflected in the pairwise comparisons, the maximum likelihood tree, and recombination corrected phylogeny (Figs 4A, 4B and S1). While the majority of isolates within the ancestral group are highly related, one isolate, SC4, is more divergent and clusters separately on the tree. The average pairwise genetic distance between SC4 and each ancestral isolate is 173 nucleotide differences whereas the average pairwise genetic distance among all other unique pairs of ancestral isolates is 39 nucleotide differences. Thus, we named two subgroups of the ancestral isolates according to this divergence (1A and 1B). Within the recent group, there is additional clustering of the isolates into two subgroups, named 2A and 2B. Group 2A, is comprised of 12 total isolates with 147 unique polymorphisms (SNPs and indels) and group 2B is comprised of 15 total isolates with 216 unique polymorphisms.

Isolates collected from the most recent time point originated from biopsy samples from distinct stomach regions, allowing us to examine if region specific adaptation drives subgroup formation. To assess this, π for all the pairs isolated from the same source biopsy (within region) was compared to π from all the pairs from different source biopsies (between region). Although we hypothesized that isolates from the same source biopsy would be more similar, we instead found between region pairs have the same level of diversity as within region pairs (Fig 4C). We also did not find any specific SNPs or indels associated with isolates from either the antrum or corpus (S5 Table). This suggests subgroup differentiation within time points is not defined by stomach region adaptation in this patient. Consistent with this finding, both recent subgroups (yr 2000) contain isolates from all three biopsy locations and nearest neighbors on the tree often come from different biopsies (Fig 4A). The formation of distinct subgroups within a population of isolates collected from a single time point, suggests the

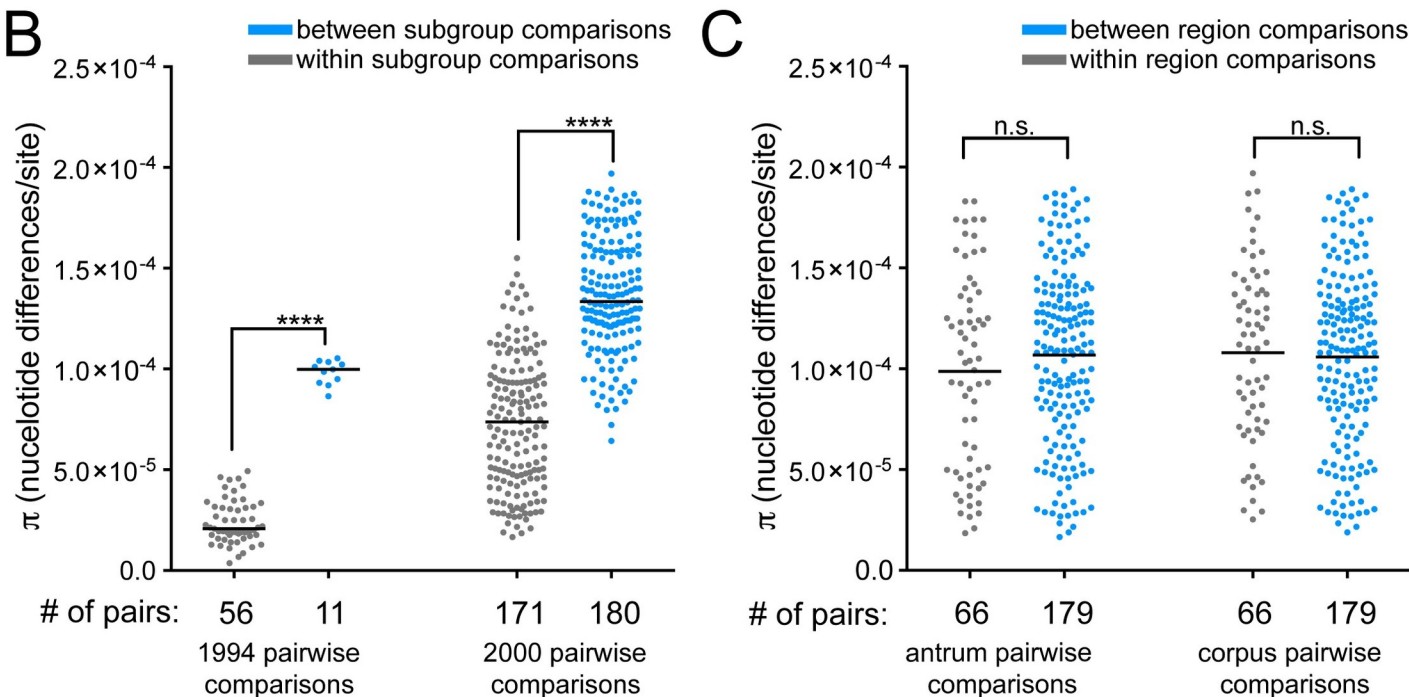

**Fig 4. Clustering of strains by genetic similarity suggests distinct subgroups that do not correlate with biopsy site.** (**A**) A maximum likelihood tree was generated from all SNPs in the collection with Nextstrain [55]. Isolates were named according to the anatomic region of their source biopsy as annotated in the key and branch coloring was added to distinguish genetically related subgroups within the collection. Light and dark blue isolates are from 1994 (1A, 1B, respectively). Yellow (2A) and

red isolates (2B) are from 2000. The X-axis shows the divergence from the ancestral root (number of substitutions (subs)/site).(**B-C**) Each point on the plot represents the pairwise genetic distance calculated for groups of isolates described with a black bar representing the mean ($\pi$). (**B**) Pairwise comparisons of isolates within subgroups displayed on the maximum likelihood tree (gray) have smaller genetic distances on average than pairwise comparisons of isolates from different subgroups (blue) from the same time point. Significance was determined using a Student's t-test (****, p<0.0001). (**C**) Pairwise comparisons of recent isolates within the same stomach regions (gray) have the same average genetic distance as pairwise comparisons of isolates from different stomach regions (blue) in both the antrum and corpus.

possibility of niche level adaptation, but these distinct niches must be present in all regions of the stomach sampled.

## Recent *H. pylori* isolates have increased proinflammatory activity driven by *cagY* genetic variation

Substantial genetic divergence of *H. pylori* populations over this six year period of infection, coupled with enrichment of polymorphisms in genes related to virulence, prompted exploration of pathogenic phenotypes. First, we tested ability of each strain to initiate an inflammatory response. Each of the 39 isolates was co-cultured with a gastric epithelial cell line (AGS) for 24hrs (MOI = 10) and the release of inflammatory cytokine interleukin-8 (IL-8) was measured in the supernatants. The J99 ancestral strain and J99 *ΔcagE*, a mutant that blocks assembly of the Cag T4SS, were used as controls in each independent experiment. All isolates were Cag PAI+ and induced IL-8 at levels above J99 *ΔcagE*. However, isolates from the most recent time point on average induced more IL-8 compared to ancestral isolates (Fig 5A and 5B). There was some heterogeneity in this phenotype with the least inflammatory isolates inducing 12% less and the most inflammatory isolates inducing 56% more IL-8 than J99 (Figs 5B and S3). Isolates with similar IL-8 phenotypes clustered together on the maximum likelihood tree; and comparison of the genetic and phenotypic distances between pairs of isolates from both time points (n = 276) showed genetic divergence correlates with phenotypic divergence in induction of IL-8 (Fig 5C). There is also a weaker correlation between genetic and phenotypic distances of within timepoint pairs, indicating genetic divergence within the population at a single time point may also contribute to the phenotypic heterogeneity observed (Fig 5D). These data show that isolates able to induce more inflammation persisted, indicating a possible adaptive advantage of pro-inflammatory activity during chronic infection in this patient.

To investigate the genetic basis of shared IL-8 phenotypes, we focused on nSNPs that occurred within the Cag PAI. While there was no enrichment of nSNPs within the Cag PAI as a whole (chi-squared, p-value = 0.3921), we did see enrichment in two Cag PAI genes, *cagY* and *cagA* (S4A Fig and S3 Table). Recent isolates had 7 unique nSNPs in *cagA*, however none were localized to known functional domains (S4B Fig) [57]. Several nSNPs were detected within the middle repeat region of *cagY*. This domain contains a series of long and short direct repeat sequences that can undergo recombination resulting in expansion or contraction of repeats. This can attenuate or enhance Cag T4SS-dependent IL-8 secretion. In animal models of infection, recombination events that diminish pro-inflammatory activity are dependent on adaptive immunity [15]. Due to the difficulties in precisely mapping these recombination events with short-read WGS data, we utilized restriction fragment length polymorphism (RFLP) together with Sanger sequencing to identify unique alleles of *cagY* within the collection (Figs 6A, 6B and S5C). All of the recent isolates (Groups 2A and 2B) and ancestral isolate SC4 (Group 1B) share the same RFLP pattern, which is distinct from the RFLP pattern shared by the other ancestral isolates (Figs 6A and S5A). Sanger sequencing revealed that all the isolates in group 1A, including J99, share the same sequence. However the allelic variant of *cagY* in SC4 is distinct from that in the recent isolates (S5B Fig). The SC4 *cagY* allele carries two nSNPs shared with recent isolates, including one that introduced a restriction site seen by RFLP,

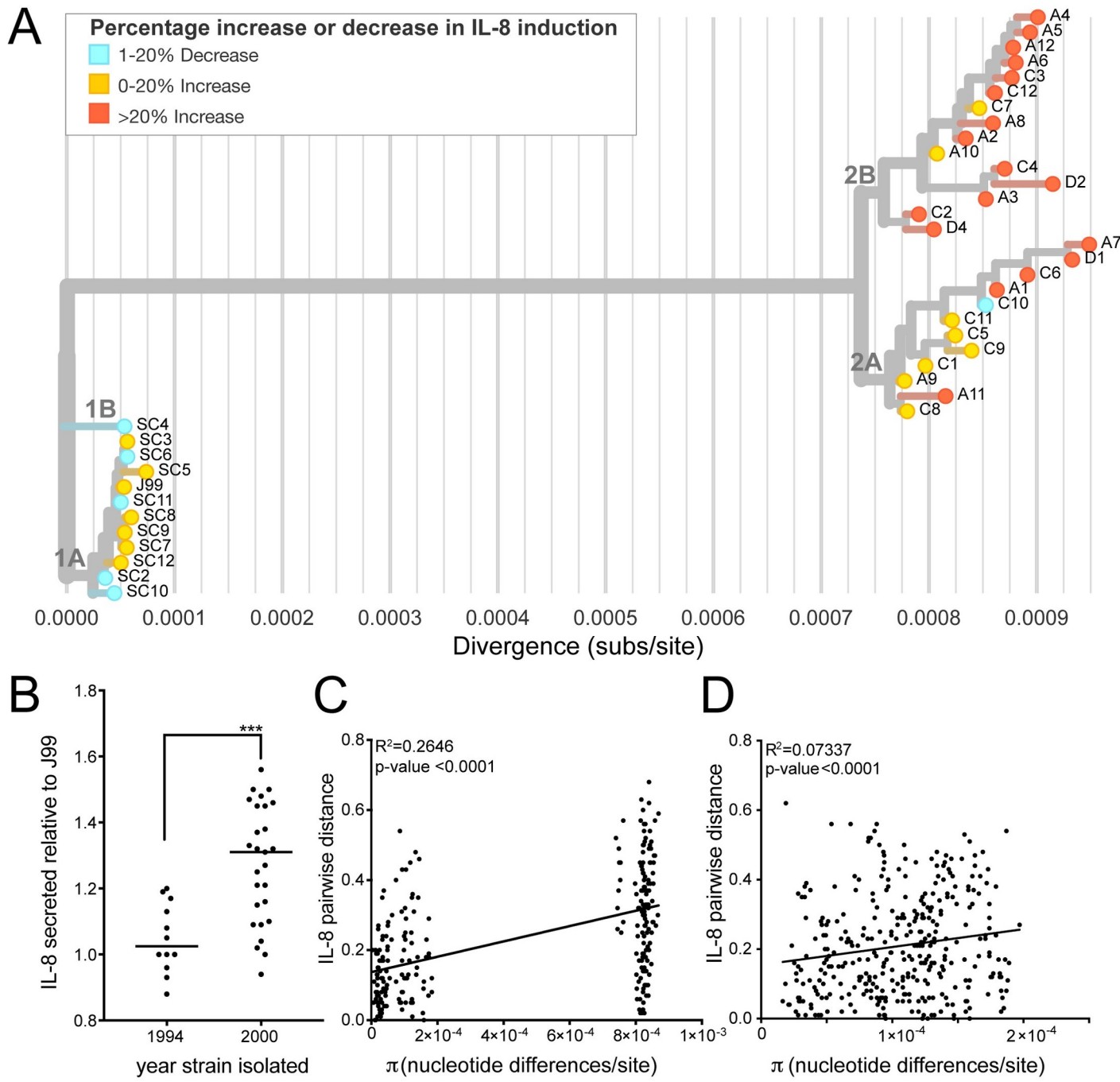

**Fig 5. Isolates from recent time point have increased induction of IL-8 secretion during co-culture with gastric epithelial cells.** (**A**) The maximum likelihood tree from Fig 4A overlaid with IL-8 induction phenotype of each isolate after 24 hours of co-culture (MOI = 10) with gastric epithelial cell line (AGS). Leaf colors represent percent increased or decreased IL-8 secretion relative to ancestral isolate J99. (**B**) Each point shows mean value of IL-8 detected in the supernatants of infected AGS cells relative to J99 for independent isolates. The mean value was calculated from at least two experiments with triplicate wells. Black line represents the mean values from each subset of isolates (yr 1994, yr 2000). Significance was determined using a Student's t-test (****, p<0.0001). (**C-D**) Comparisons of genetic (π) and phenotypic (relative IL-8 secreted) distances between unique pairs of isolates from different time points (**C**) and within the same time point (**D**) are shown. Plot shows a linear regression with p-value derived from F-test and correlation coefficient ($R^2$) reported.

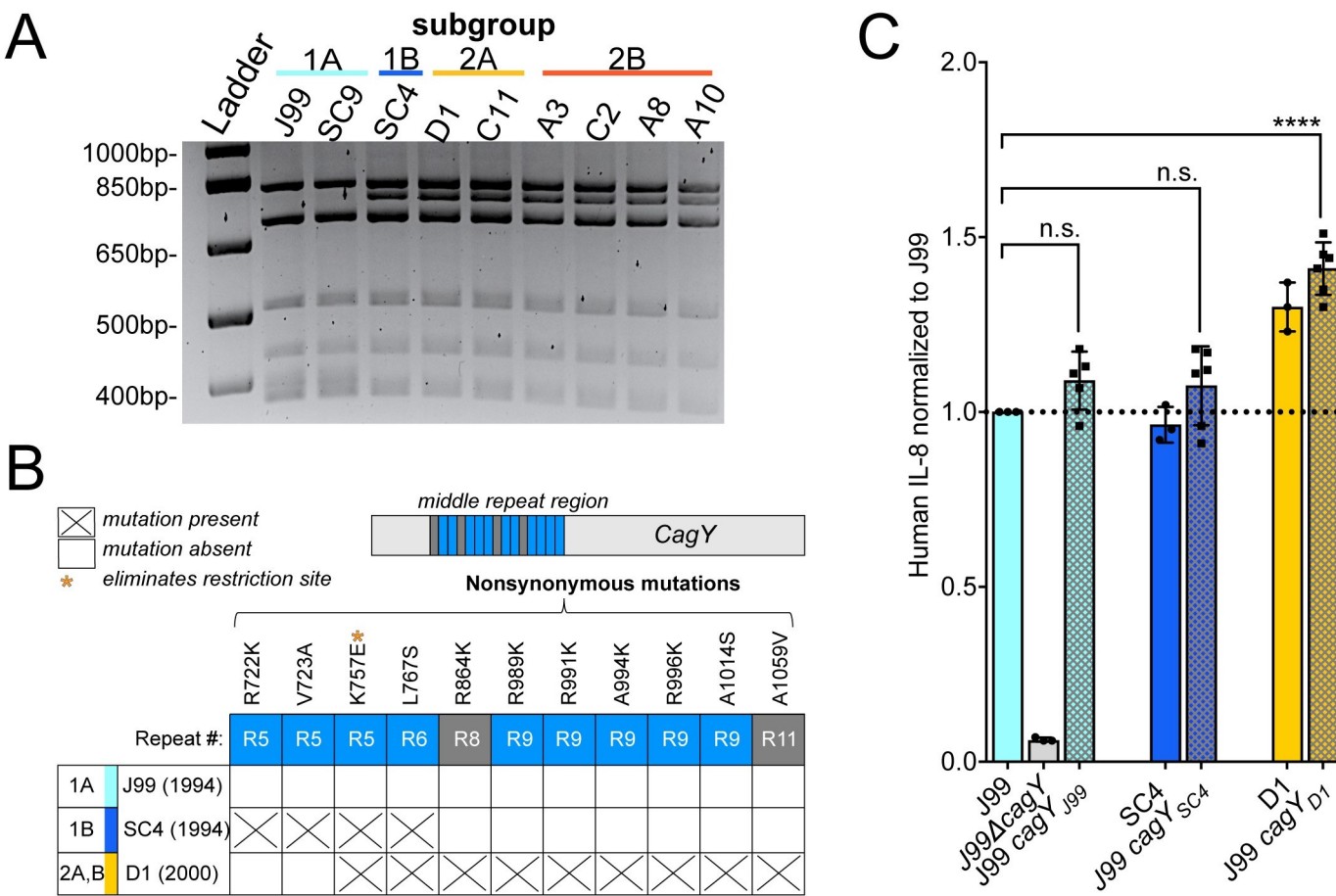

**Fig 6.** *cagY* **genetic polymorphisms promote enhanced IL-8 secretion.** (**A**) RFLP analysis of amplified *cagY* repeat region from representative isolates digested with restriction enzyme (DdeI) reveals two distinct patterns within the populations. Isolates are colored by subgroup as in Fig 4A. (**B**) Amino acid polymorphisms for the three different allelic variants of *cagY* in our isolate collection detected by Sanger sequencing. Isolates listed represent the three alleles (J99, SC4, D1) with date of isolation and subgroup(s) indicated. All nSNPs detected map within the middle repeat region of *cagY* (15 total repeats; short in blue and long in gray). (**C**) Levels of IL-8 produced by *cagY* allelic exchange strains relative to J99 ancestral (y-axis line = 1) 24hrs post infection of AGS cells (MOI = 10). Data points represent averaged values from triplicate wells from at least 3 independent experiments. Significance was determined with a one-way ANOVA with Dunnett's corrections (n.s. not significant, **** p<0.0001).

however it also harbors two unique nSNPs not found in any other isolates in the collection. All recent isolates (Groups 2A and 2B) have 9 nSNPs total compared to the J99 *cagY* allele including the two shared with SC4 (Figs 6B and S5C). None of the alleles had expansion or contraction of the number of repeats, but likely arose from gene conversion from sequences within other repeats (S5C Fig). In order to test for a functional link between the variation in *cagY* and the differences in IL-8 phenotype, we performed an allelic exchange experiment, replacing the *cagY* allele in J99 ancestral strain with the two other *cagY* allelic variants (Fig 6C). Co-culture of these engineered strains with AGS cells showed that the *cagY* allele shared by the recent isolates confers the increase in induction of IL-8 at 24hrs. The SC4 allele in the J99 genomic context induced similar levels of IL-8 secretion as J99. Therefore modulation of T4SS function can occur through the introduction of specific SNPs in the absence of expansion or contraction of the *cagY* repeats. The same experiment was performed with *cagA* variants, but IL-8 induction was not significantly different from J99 (S4C Fig).

## Sub-populations within the collection have distinct bacterial cell morphologies

Cell morphology has also been linked to virulence in *H. pylori* [58]. In order to measure cell morphology we used CellTool, a program which takes 2-D phase contrast images and measures quantitative cell shape parameters from cell outlines [58]. Based on these measurements, isolates were divided into three phenotypic shape categories—short pitch, long pitch, and rod. Short pitch isolates have increased wavenumber per unit centerline axis length compared to the long pitch and rod isolates. Rod isolates have decreased side curvature per unit centerline axis length compared to the short and long pitch isolates (Fig 7B and 7C). Isolates with similar shape phenotypes cluster on the maximum likelihood tree (Figs 7A and S6). Interestingly, all ten rod-shaped isolates from group 2B had loss of function mutations in *H. pylori* cell shape determining (*csd*) genes known to cause rod-shape morphology when deleted. We observed four unique mutations in *csd4*, one unique mutation in *csd5*, and three unique mutations in *csd6*, one of which was shared by three isolates (S7 Fig). Thus, group 2B appears to have convergent evolution leading to straight-rod morphology (Fig 7D).

Pairwise comparisons show that recent isolates within the same cell shape phenotype category are more genetically similar than recent isolates from different cell shape categories (Fig 8A). Additionally, plots of the genetic and phenotypic distances between unique pairs of recent isolates from both time points (n = 351) showed that genetic divergence positively correlates with phenotypic divergence both in wavenumber and size curvature per unit axis length (Fig 8B and 8C). This indicates a signature of selection for loss in helical shape within this sub-population of recent isolates.

## Isolates differ in mouse colonization during acute infection

Considering the observed phenotypic divergence among isolates between and within time points, we hypothesized that individual isolates may behave differently in a mouse stomach colonization model. C57BL/6 mice were infected with representative isolates from each time point and subgroup for 1 week. All isolates tested successfully colonized mice. However, the proportion of mice with detectable infection and loads (CFU/gram of stomach tissue) differed. Almost all the mice infected with the two isolates from group 2A (C11, D1) and the single isolate from the ancestral group 1B (SC4) had higher loads than representative isolates from the other groups and this increase coincided with a greater proportion of mice stably infected after one week compared to the others (Fig 9A). Isolates within different cell shape categories and IL-8 profiles were chosen when possible. Although loss of helical shape has been shown to decrease colonization, both helical and rod-shaped isolates from clade 2B infected at lower loads (Fig 9C). Since recombination events in *cagY* were detected in all isolates with increased loads, and these changes impacted the inflammatory response in-vitro, we tested to see if our ancestral strain (J99) with the recent variant of *cagY* (J99 *cagY*$_{D1}$) would also colonize at higher loads. However, J99 *cagY*$_{D1}$ colonized mice similarly to J99 ancestral (Fig 9B), indicating the increased mouse colonization phenotype is not conferred by *cagY* variation. Together these results suggest that there are additional, unknown properties of these isolates contributing to colonization (Fig 9C). However, the differences in colonization between groups 2A and 2B and 1A and 1B supports the assertion that subgroup differentiation has phenotypic consequences for infection.

## Discussion

Within-host *H. pylori* isolates from a single individual, once thought to be homogenous, have since been shown to be genetically distinct. Next-generation sequencing provides tools to

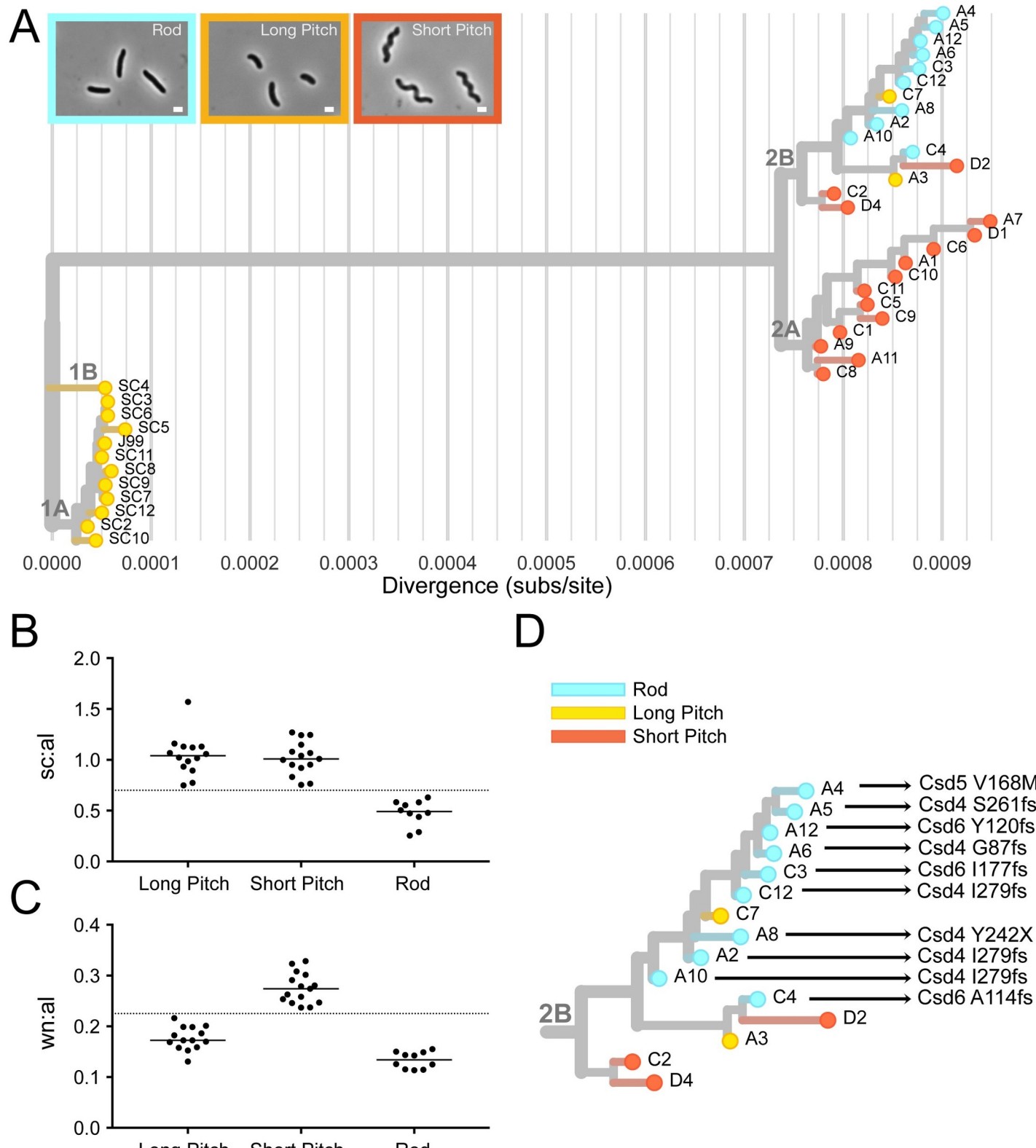

**Fig 7. Cell morphology varies among genetically distinct subgroups.** (**A**) Cell shape phenotype clustering on maximum likelihood tree. Leaves colored to indicate cell morphology phenotype of each isolate with rods in blue, short pitched in red, and long pitched in yellow. Representative phase contrast micrographs of each morphologic class shown. (magnification = 100x, scale bar = 1 μm). (**B-C**) Cell shape parameters calculated from 2-D phase images with CellTool software for isolates

with indicated cell morphologies. Individual points represent mean values for measurements taken from >100 cells/isolate. Side curvature and wavenumber values were normalized by cell centerline axis length. (**B**) Mean side curvature values normalized by centerline axis length (sc:al) is decreased in rod shaped cells (<0.7, as indicated by y-axis line) and (**C**) wavenumber normalized by centerline axis length (wn:al) is increased for cells that have increased wavenumber (>0.225, as indicated by y-axis line). (**D**) Subgroup 2B labeled with amino acid mutations in cell shape determining proteins (Csd4, Csd5, Csd6).

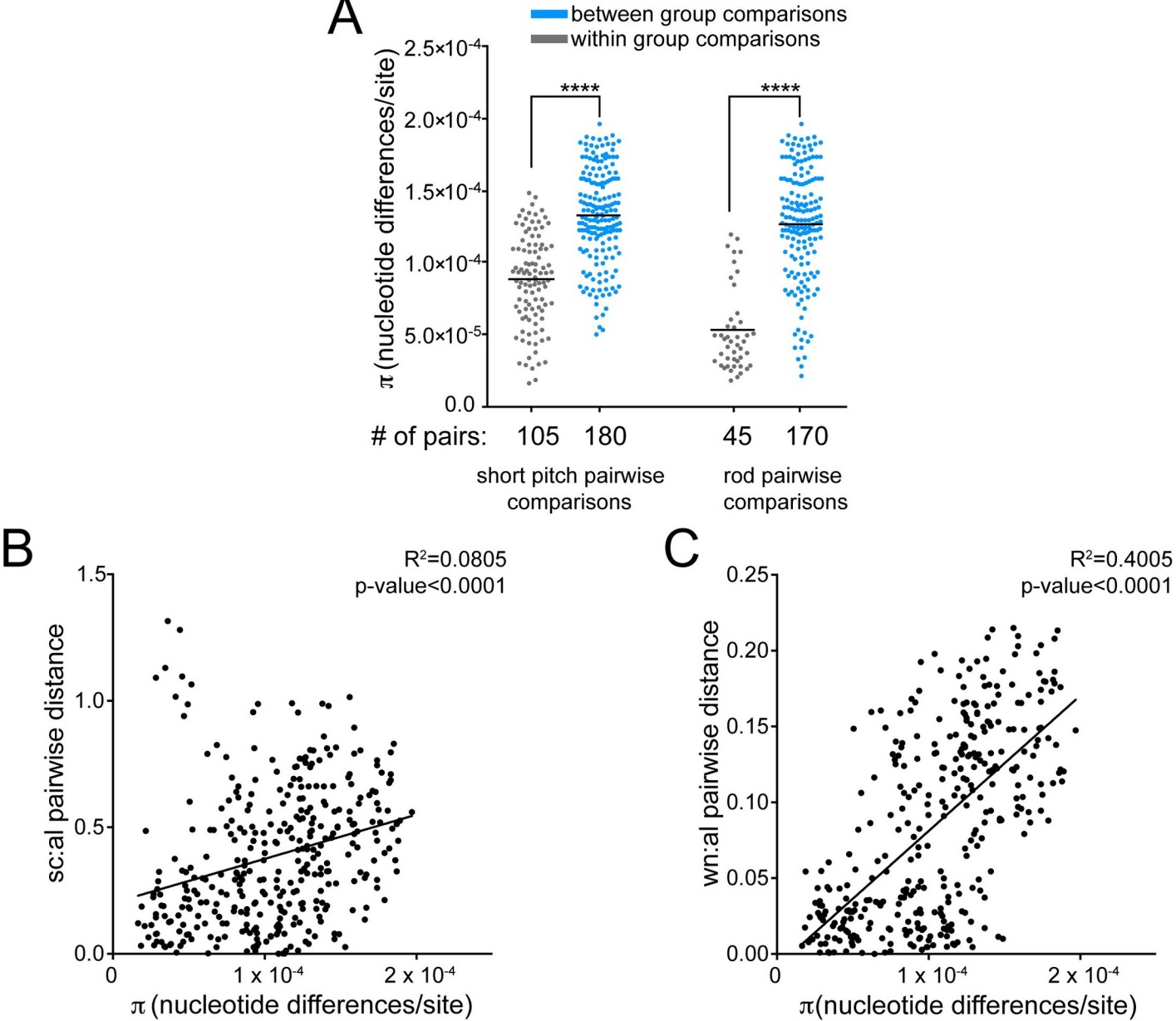

**Fig 8. Cell morphology parameter divergences correlate with genetic distance during chronic infection.** (**A**) Pairwise comparisons show that recent isolates within the same cell shape phenotype category (gray) are more genetically similar than recent isolates from different cell shape categories (blue). Each point represents a unique pairwise comparison between recent isolates. Midline represents the mean (π statistic). Significance was determined using Student's t-test (**** $p<0.0001$). (**B-C**). Correlation of genetic and phenotypic distances in (**B**) side curvature (sc) and (**C**) wavenumber (wn) per unit centerline axis length (al) between unique pairs of recent isolates (n = 351) with each point displaying a unique pairwise comparison. Plot shows a linear regression with p-value derived from F-test and correlation coefficient ($R^2$) shown.

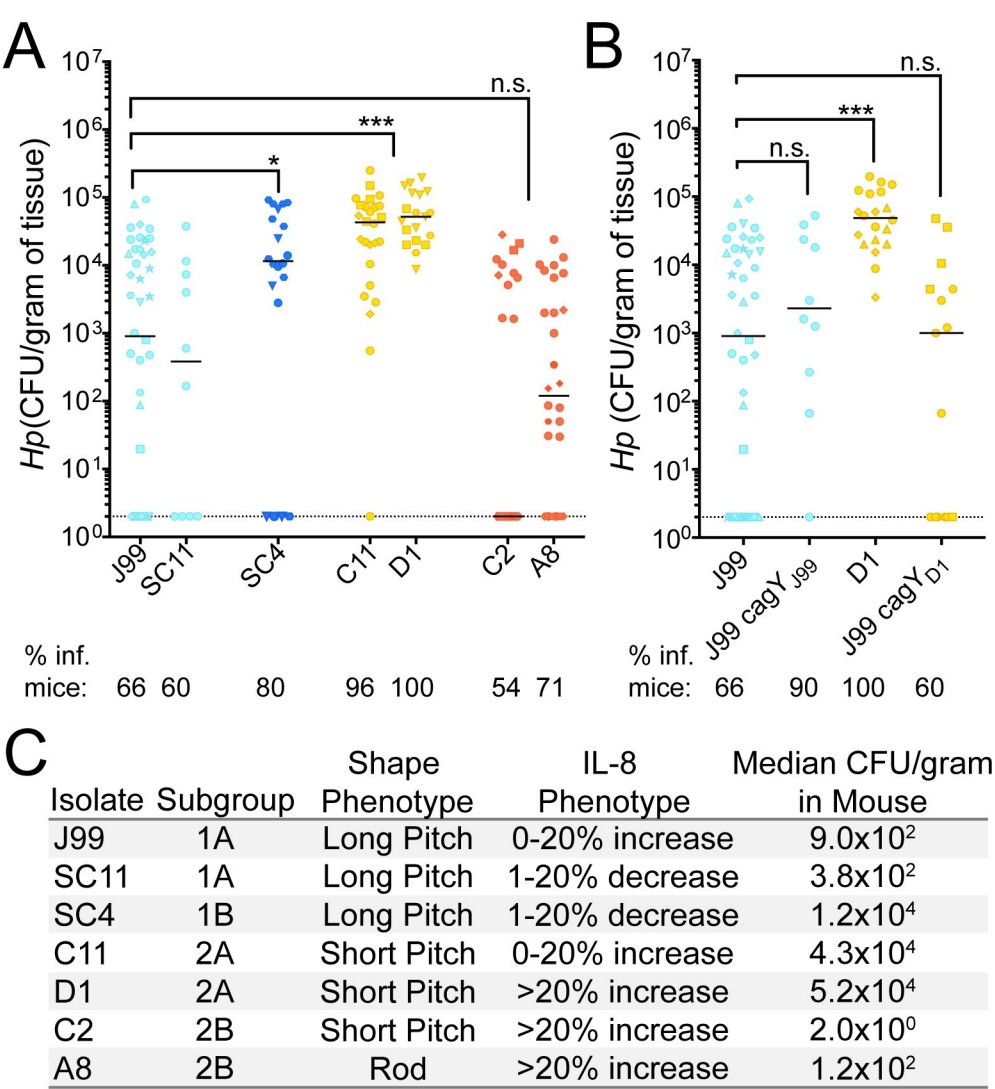

**Fig 9. Mouse colonization among isolates from distinct subgroups differs and is not explained by *cagY* variation.** (**A-B**) Each point represents the colony forming units per gram of mouse tissue homogenate from a single mouse. Two CFU/gram is the limit of detection (dotted line). Biological replicates indicated by different symbol shapes with the median are shown. P-values were calculated from pooled experimental replicates using a Mann-Whitney non-parametric test (n.s. not significant, * p<0.05, *** p<0.001). Percentage of mice with bacterial colonization above the limit of detection (% inf. mice) is indicated below the isolate names. (**A**) Colonization in WT B6 mice 1 week post infection with representative isolates from each subgroup. Color indicates maximum likelihood tree subgroup 1A (light blue), 1B (dark blue), 2A (yellow), and 2B (red). (**B**) Alleles of *cagY* from indicated strains were engineered into strain J99 at the native locus and resultant isolates used for infection experiments. Data points in blue are mice infected with J99 variant of *cagY* and points colored in yellow were infected with D1 variant of *cagY*. (**C**) Summary of representative isolate phenotypic characteristics including the subgroup, bacterial shape, IL-8 phenotype (relative IL-8), and median mouse colonization (CFU/gram).

examine the breadth of diversity present, however little is known about how this diversity contributes to pathogenesis and disease progression. Our study characterized both genetic and phenotypic diversity of infecting populations from a single, chronically infected individual at two time points over a six-year period. We re-sequenced isolate J99, originally isolated from the patient in 1994 and detected 755 polymorphisms differing from the published reference (GenBank, AE001439). The majority (553 polymorphisms) were shared across all isolates, and

were therefore excluded from the analysis. These polymorphisms reflect either improvements to sequencing methods since the original sequence was published in 1999, differences in the J99 stock used in our experiment compared to that of Alm et al. (AE001439), or misalignments of short read data to the reference.

Previous studies have found within host evolution of *H. pylori* is shaped both by de-novo mutation and homologous recombination events, with recombination events generating the majority of the overall diversity [59]. Polymorphism introduced by both mechanisms can impact phenotype, therefore with our analysis we reported and explored sequence level diversity acquired by both mechanisms. Others have estimated the within-host mutation rate to be between 6.5–0.5 x $10^{-5}$ substitutions per site per year by excluding predicted recombination sites in order to make evolutionary inferences [35, 44, 47, 53]. However, it remains debated whether diversity generated via recombination versus mutation can be accurately identified and filtered to reconstruct evolutionary relationships [60]. Since clustered nucleotide polymorphisms (CNPs), typically excluded from molecular clock rate calculations, were included in this analysis, our overall within-host molecular clock rate of 1.3 x $10^{-4}$ subs/site/yr is elevated compared with rates based on substitutions from mutation alone [44]. However, when we excluded polymorphism introduced via importation events using ClonalFrameML, we calculated a molecular clock rate of 2.2 x $10^{-5}$ subs/site/yr, consistent with the other published estimates [44, 47]. This analysis also upholds a model of within host diversity arising predominantly from recombination.

We found that in this individual, with no known exposure to antibiotics, infecting populations increased diversity over time and clustered into genetically distinct subgroups, suggesting adaptation to specific host niches [34]. The high number of polymorphic sites within OMPs, including *hopQ*, *sabA*, *babA*, and *sabB*, were predicted to have accumulated from recombination events and likely originated from a genetically related co-infecting strain, which was either not sampled or not present in the original population. High variation in these genes is unsurprising considering paralogous OMPs have been shown to diversify within-host via frequent intra and intergenomic recombination [22]. Accumulation of nSNPs in OMPs supports a model of adaptation driven by interactions with the host environment. In addition to OMPs, many genes that accumulated an excess of nSNPs had unknown function, providing additional targets to examine in future studies.

In other chronic infections, such as *Pseudomonas aeruginosa* infection in cystic fibrosis patients, the emergence of sub-populations is driven by region specific adaptation within distinct anatomical regions of the lung [61]. Evidence of anatomical stomach region specific adaptation in *H. pylori* infections is limited, but it appears to occur in only a small subset of patients [37].These signatures may be obfuscated by frequent population mixing and migration or deterioration of structured niches due to loss of acid production and other tissue changes [42].

While our data do not support subgroup divergence by anatomic region in this individual, the selective pressures at play appear to correspond to known pathogenicity phenotypes. Nonsynonymous polymorphisms detected only within the recent population (yr 2000), fell within known virulence genes, including OMPs involved in host adhesion and Cag PAI-associated genes and distinct alleles within Cag PAI were confirmed by Sanger sequencing. Differences in IL-8 secretion, bacterial cell morphology, and ability to colonize a mouse in an acute infection model were discovered among isolates, suggesting subgroup divergence driven by tissue features that vary in the stomach across multiple anatomic locations.

Recombination within the middle repeat region of *cagY*, resulting in expansion or contraction of repeats, occurs frequently in short-term animal infections and in humans [15]. We observed modulation of IL-8 induction mediated by T4SS function through mutation and/or

recombination without expansion or contraction of *cagY* repeats. The finding that isolates at later time points were more pro-inflammatory was surprising considering inflammation is thought to limit bacterial burden. However, *H. pylori* persists despite relatively high levels of inflammation, so it is possible this feature may be exploited during chronic infection in order to reduce competition for host resources by members of the microbiota [62].

Isolates within this collection also had differences in cell morphology. Morphology differences have been observed among strains from different individuals [63]; here we find that *H. pylori* morphologies differ among isolates from a single patient. In subgroup 2B, we discovered convergent loss of helical cell shape through multiple unique frameshift mutations in cell shape determining (*csd*) genes. Rod-shape isolates have previously been shown to have a colonization deficit manifest at early time points, but to recover during 1–3 months of chronic infection in mice [64]. Due to clustering of rod shapes in subgroup 2B, we suspect that helical shape, while important for early infection and transmission, may be detrimental at later stages of human infection or in particular stomach niches. Among isolates that retained helical shape, we detected more subtle differences in helical pitch, but it is unknown what genetic determinants are responsible or if these differences have direct impacts on colonization.

The observed differences in mouse colonization between isolates from each sub-population supports our initial hypothesis that there are functional consequences of sub-population divergence. Typically, clinical isolates infect mice poorly as mice are not natural hosts for *H. pylori*. However, a few clinical isolates have the intrinsic ability to colonize and can become more robust via serial passage in the mouse stomach [65]. Bacterial properties, including chemotaxis, cell shape, and activity of the Cag PAI, impact mouse colonization and are likely important in establishing human infections [66]. In our collection, there was heterogeneity in mouse colonization among isolates that corresponded to subgroup. Robust colonizers may be more likely to be involved in person-to-person transmission in humans, but it is also possible that these strains may behave differently in other animal models or human hosts. Increases in colonization potential of representative isolates within 1B and 2B sub-populations does not correlate with differences in morphology or IL-8 phenotypes, indicating an additional unknown factor or combination of factors is responsible for conferring a colonization advantage. Further exploration of the genetic basis for mouse colonization advantage using the subgroup specific variants defined in this study may give new clues to the complex selective forces operant during chronic stomach colonization by *H. pylori*.

## Materials and methods

### Ethics statement

All procedures were approved by Vanderbilt University and Nashville Department of Veterans Affairs institutional review boards. All mouse experiments were performed in accordance with the recommendations in the National Institutes of Health Guide for the Care and Use of Laboratory Animals. The Fred Hutchinson Cancer Research Center is fully accredited by the Association for Assessment and Accreditation of Laboratory Animal Care and complies with the United States Department of Agriculture, Public Health Service, Washington State, and local area animal welfare regulations. Experiments were approved by the Fred Hutch Institutional Animal Care and Use Committee, protocol number 1531.

### Growth and isolation of *H. pylori*

In the initial sampling, a total of 43 *H. pylori* isolates (13 (antral, 1994), 5 (duodenum, 2000), 12 (corpus, 2000), 1 (cardia), and 12 (antral, 2000)) were collected from biopsy samples from two separate upper gastrointestinal endoscopies performed in a single 48-yr old Caucasian

male (1994) residing in Tennessee and treated at the Nashville VA Medical Center. A total of 39 isolates with sufficient sequence coverage (30x) were analyzed in this study (Fig 1). *H. pylori* isolates were grown on solid media, horse blood agar (HB agar) or shaking liquid cultures. HB agar plates contain 4% Columbia agar base (Oxoid, Hampshire, UK), 5% defibrinated horse blood (Hemostat Labs, Dixon, CA), 10 mg/ml vancomycin (Thermo Fisher Scientific, Waltham, MA), 2.5 U/ml polymyxin B (Sigma-Aldrich, St.Louis, MO), 8 mg/ml amphotericin B (Sigma-Aldrich), and 0.2% β-cylodextrin (Thermo Fisher). For HB agar plates used to grow *H. pylori* from homogenized mouse stomach, 5 mg/ml cefsulodin (Thermo Fisher), 5 mg/ml trimethoprim (Sigma) and 0.2mg/uL of Bacitracin (Acros Organics, Fisher) are added to prevent outgrowth of mouse microflora. Shaking liquid cultures were grown in brucella broth (Thermo Fisher Scientific, Waltham, MA) supplemented with 10% heat inactivated FBS (Gemini BioProducts, West Sacramento, CA). Plates and flasks were incubated at 37˚C under micro-aerobic conditions in 10% $CO_2$, 10% $O_2$, 80% $N_2$, as previously described [67]. For resistance marker selection, HB agar plates were supplemented with 15 μg/ml chloramphenicol, or 30 mg/ml sucrose, as appropriate.

## DNA extraction, genome sequencing

Genomic DNA from each isolate to be sequenced was purified using the Wizard Genomic DNA Purification Kit (Promega, Fitchburg, WI) and libraries were constructed and indexed using Nextera$^{RTM}$ DNA Library Prep Kit (Illumina, San Diego, CA) and Nextera$^{RTM}$ Index Kit (Illumina). All cultured isolates (n = 43) were sequenced on an Illumina MiSeq instrument in the Fred Hutchinson Cancer Research Center Genomics Shared Resource. Using J99 ancestral as the reference strain (AE001439), variants were called from raw paired end reads using the Breseq v0.35.0 software with default parameters and SNPs were further validated using default Samtools software suite [68]. Four isolates with average coverage below 30x were dropped from the analysis. Of the remaining 39 isolates, 34 had coverage >100x. Additional sequencing metrics can be found in S6 Table. Short read fastq sequence files from the remaining 39 isolates in this study are publicly available on NCBI SRA database (BioProject accession: PRJNA633860, <https://www.ncbi.nlm.nih.gov/sra/PRJNA622860>).

## Enrichment of nonsynonymous SNPs in genes and functional gene classes

Gene annotations were made using the available Genbank file available for J99 (AE001439) with some manually added annotations of OMPs. All annotation files are available at <https://github.com/salama-lab/Hp-J99>. For identification of genes with excess accumulation of nSNPs and the number of alleles in each population, all unique nSNPs accumulated over the six years reported in Table 1 were used. Z-scores were calculated using number of counts per gene normalized according to gene length. Genes with nSNP accumulation greater or equal to four standard deviations from the mean are listed in Table 2. Z-scores for all genes are listed in S3 Table. To identify enrichment of nSNPs within functional gene-sets, each of the 1,495 genes were annotated with designations in the Microbial Genome Database (MGDB, http://mbgd.genome.ad.jp/) [50]. A Fisher's exact test was used to identify MGDB gene class categories with enrichment or depletion of nSNPs. The number of nSNPs falling within certain MGDB categories were compared to expected values based on a normal distribution and p-values were corrected for multiple testing using Benjamini and Hochberg false discovery rate methods [69]. Adjusted p-values <0.03 were considered statistically significant. To identify markers of antimicrobial resistance, polymorphisms detected in the collection were screened against a list of annotated antimicrobial resistance genes downloaded from PATRIC <https://www.patricbrc.org>.

## Pairwise distance calculations

The number of nucleotide differences per site (genetic distance) between pairs of isolates was calculated using PopGenome (R, v 4.0.2) using the merged sample variant call file (VCF) file available on Github (nucleotide diversity, π, [70], PopGenome, [71]). All sites that did not align to the reference genome, J99, or had read depth <5 were excluded from the analysis, which removed missing or ambiguous sites resulting from structural variation or mis-mapping. The unique number of shared sites for each pair was calculated using the BEDtools intersect function (reported in S7 Table) [72]. The total number of nucleotide differences for each pair was derived from the merged sample VCF file with low quality sites filtered according to the read depth parameters above. All detected indels were excluded from this analysis to avoid inflation of genetic distance due to alignment errors within highly repetitive regions [73]. The statistical significance of differences between groups was assessed using Student's t-test as indicated in figure legends. The merged sample VCF file is available on the Github page (https://github.com/salama-lab/Hp-J99/blob/master/data/Hp_J99_032020.vcf.gz).

## Construction of the maximum likelihood and recombination-corrected trees

SNPs detected across each isolate reported in the merged sample VCF were imported into the reference J99 (AE001439) to create a consensus genome for each of the 39 isolates. The maximum-likelihood tree (Figs 4A, 5A and 7A) was built in IQtree implemented in Nextstrain v 1.8.1, using a general time reversible (GTR) substitution model. The ancestral root was inferred using joint maximum likelihood algorithm. All datasets, config files, documentation, and scripts used in this analysis or to generate figures are available publicly on our Github page <https://github.com/salama-lab/Hp-J99>. The interactive tree can be found at the Nextstrain community site <https://nextstrain.org/community/salama-lab/Hp-J99>. To create the recombination corrected tree with rescaled branch lengths, the reference-aligned consensus genomes were aligned using progressiveMauve assuming collinear genomes and a new rescaled tree (S1B Fig) was generated using ClonalFrameML using all sites in the reference (AE001439) [53, 74]. Time scaled trees and molecular clock rates (S2 Fig) were generated with TreeTime implemented within Nextstrain, which approximates divergence times and molecular clock rates using maximum likelihood methods [75]. Branch lengths of the maximum likelihood and recombination corrected trees were compared using the ape and adephylo packages in R, v 4.0.2.

## Sequencing and PCR-RFLP of *cagY* middle repeat region

The *cagY* sequences were determined using Sanger sequencing using primers listed in S8 Table and PCR-RFLP as previously described [15]. Flanking primers were used to amplify the *cagY* repeat region from every isolate. Amplicons were purified with QIAquick PCR purification kit according to the instructions from the manufacturer (Qiagen, MD) and digested with restriction enzyme DdeI (New England Biolabs, Ipswich, MA). Digested amplicons were run on a 3% agarose for visualization after ethidium bromide staining.

## Construction of *H. pylori* mutants

Six J99 mutants were constructed (J99Δ*cagY*, J99 *cagY*$_{D1}$, J99 *cagY*$_{SC4}$, J99 Δ*cagA*, J99 *cagA*$_{D1}$, J99 Δ*cagE*) and are listed in S9 Table. Isogenic knockout mutants, J99 Δ*cagY* and J99 Δ*cagA*, were constructed using a vector-free allelic replacement strategy. Upstream and downstream genomic regions flanking the gene were amplified and ligated to a *catsacB* cassette, which

confers mutants both chloramphenicol resistant (*cat*) and sucrose sensitivity (*sacB*). Positive clones were selected with 15 μg/ml chloramphenicol, as previously described [76, 77]. We integrated variant alleles of the deleted gene at the native locus using sucrose counter selection. All mutants were validated via diagnostic PCR and Sanger sequence. Primers used for generating *H. pylori* mutants are listed in S8 Table in the supplemental material.

### *H. pylori* co-culture experiments and IL-8 Detection

AGS cells, from a human gastric adenocarcinoma cell line (ATCC CRL-1739), were grown in Dulbecco's modified Eagle's medium (DMEM) (Thermo-Fisher) supplemented with 10% heat-inactivated FBS (Gemini-Benchmark). For co-culture with *H. pylori*, AGS cells were seeded at $1 \times 10^5$ cells/well in 24-well plates 16 h prior to infection. The day of infection, medium was removed from AGS cells and mid-log-phase (optical density at 600 nm (OD) 0.3–0.6) *H. pylori* resuspended in DMEM–10% FBS–20% Brucella broth was added at multiplicity of infection of 10:1. Supernatants from triplicate wells of each condition were collected at 24 hrs and assayed for the IL-8 concentration using a human IL-8 enzyme-linked immunosorbent assay (ELISA) kit according to the instructions of the manufacturer (BioLegend, San Diego, CA). IL-8 values were reported as normalized values defined as a proportion increased or decreased compared to values obtained for J99, which was included in each experimental replicate. P-values were calculated from at least two biological replicates where each isolate was assayed with triplicate wells using a one-way ANOVA with Dunnett's corrections.

### Analysis of cell morphology

Phase contrast microscopy and quantitative analysis using CellTool software package was performed as previously described [58]. Bacterial cell masks were generated through thresholding function in ImageJ. Average side curvature, wavenumber, and centerline axis length were derived from thresholded images of bacteria (>100 cells/strain) using the CellTool software package. Average parameters were then used to calculate side curvature or wavenumber to centerline axis length ratios for each isolate.

### Mouse colonization

Female C57BL/6 mice 24–28 days old were obtained from Jackson Laboratories and certified free of endogenous *Helicobacter* by the vendor. The mice were housed in sterilized microisolator cages with irradiated rodent chow, autoclaved corn cob bedding, and acidified, reverse-osmosis purified water. All mouse colonization experiments were performed exactly as described [78]. The inoculum for each infection was $5 \times 10^7$ cells. After excision, the forestomach was removed and opened along the lesser curvature. Stomachs were divided in equal halves containing both antral and corpus regions and half stomachs were place in 0.5 mL of sterile BB10 media, weighed, and homogenized. Serial homogenate dilutions were plated on nonselective HB plates. After 5–9 days in tri-gas incubator, colony forming units (CFU) were enumerated and reported as CFU per gram of stomach tissue. P-values were calculated from pooled experimental replicates using a Mann-Whitney non-parametric test.

### Statistical analysis

Statistical analyses were performed according to tests specified above and in each figure legend using Prism v7 software (GraphPad) or R v4.0.2. P-values less than or equal to 0.05 were considered statistically significant and are marked with asterisks (*, $p<0.05$, **, $p<0.01$; ***, $p<0.001$; ****, $p<0.0001$; n.s., not significant).

## Supporting information

**S1 Fig. Comparison of maximum-likelihood tree and recombination corrected phylogeny have altered branch scale but the same overall topology.** X-axis represents divergence from common ancestor (substitutions/site) and scale bar with length indicated in red text is shown. Scale of branches is indicated in red. (**A**) The maximum likelihood tree constructed in Nextstrain ([Fig 4A](https://nextstrain.org/community/salama-lab/Hp-J99)) and (**B**) the recombination corrected phylogeny created in ClonalFrameML with rescaled branch lengths are shown. Tree construction is described in detail in the methods.
(TIF)

**S2 Fig. Molecular clock rates based on divergence from the ancestral root differ between time-scaled maximum likelihood tree and recombination corrected phylogeny.** X-axis indicates the date in years and Y-axis (**B, D**) indicates divergence (substitutions/site) from the ancestral root. (**A**) A time-scaled version of the maximum likelihood tree with (**B**) molecular clock rates inferred from the root-to-tip distance using TreeTime [75]. (**C-D**) Shows time-scaled tree (**C**) and inferred molecular clock for the recombination corrected tree (**D**).
(TIF)

**S3 Fig. Proinflammatory cytokine induction during AGS cell co-culture varies between isolates.** Maximum likelihood tree overlaid with inflammatory cytokine, IL-8, secretion phenotype after 24 hours of co-culture (MOI = 10) with gastric epithelial cell line (AGS). Leaf colors represent normalized IL-8 induction relative to ancestral isolate J99 for each isolate as shown in the figure legend.
(TIF)

**S4 Fig. Genetic variation in *cagA* does not influence IL-8 induction during AGS cell co-culture.** (**A**) Contingency table of nSNPs falling within and outside Cag PAI compared to expected values based on a normal distribution. Significance was determined using a Fisher's exact test.(**B**) CagA gene schematic labeled with nonsynonymous amino acid changes shared by all recent isolates (black bars). The three protein domains identified in the published crystal structure (blue), including the flexible N-terminal region (Domain I, amino acids 1–299), the anti-parallel beta sheet (Domain II, amino acids 304–641), and the N-terminal binding sequence (Domain III, amino acids 304–641) are labeled. Known host protein interaction motifs including the integrin binding phosphotidylserine domain, phosphotyrosine EPIYA sites, and multimerization sequence are also labeled in orange [79]. (**C**) Levels of IL-8 produced by *cagA* allelic exchange strains relative to J99 24 hrs post infection of AGS cells (MOI = 10). Data points represent averaged values from triplicate wells from at least 3 independent biological replicates. Significance was determined with a one-way ANOVA with Dunnett's corrections (n.s., not significant; **** $p < 0.0001$).
(TIF)

**S5 Fig. Three different *cagY* alleles distinguish isolate groups and subgroups.** (**A**) Maximum likelihood tree overlaid with two different *cagY* RFLP subtypes detected with restriction enzyme DdeI. RFLP subtypes, named A and B according to the figure legend, are shown in [Fig 6A](). (**B**) Maximum likelihood tree overlaid with unique *cagY* alleles detected with Sanger sequencing. Leaf colors correspond to each of the three unique alleles detected and reported in [Fig 6B](). Group 1A shares allele A, group 1B shares allele B, and groups 2A and 2B share allele C. (**C**) Amino acid alignment of multiple repeat regions of three representative *cagY* alleles detected in the collection. J99 represents the allele found in subgroup 1A (allele A), SC4 represents the allele found in subgroup 1B (allele B), and D1 represents the allele found in 2A and

2B (allele C). Polymorphic sites are highlighted with amino acids in blue representing the reference (J99, AE001439) and red indicating a nonsynonymous substitution.
(TIF)

**S6 Fig. Cell shape parameters vary within and between isolate subgroups.** Maximum likelihood tree overlaid with cell shape measurements taken from 2-D phase contrast images using CellTool. Leaf colors represent side curvature normalized by centerline axis length (**A**) or wave number normalized by centerline axis length (**B**) as indicated in the figure legends.
(TIF)

**S7 Fig. Mutations in *csd* genes are confined to isolate subgroup 2B.** Maximum likelihood tree labeled with putative loss of function mutations in cell shape determining genes (csd). Leaf colors indicate mutations in *csd4* (light blue), *csd5* (yellow), *csd6* (dark blue) listed in the figure legend with amino acid mutations. All isolates that have retained helical shape are in gray.
(TIF)

**S1 Table. A total of 1,767 unique SNPs and 485 indels were detected in the collection of 39 isolates.** For each of the 1,767 unique SNPs (**A**) and 485 indels (**B**), the tables indicate the nucleotide (nt) position and gene ID according to the reference (J99, AE001439). Unique events are labeled as either coding or intergenic. SNPs within coding regions are further subdivided into nonsynonymous or synonymous categories. The presence or absence of each polymorphism across each individual isolate is designated as present (1) or absent (0) along with the total number of isolates with the polymorphism (n).
(XLSX)

**S2 Table. Cell Envelope proteins have excess accumulation of nSNPs.** Contingency table of nSNPs falling within and outside each of the 14 MGDB class categories compared to expected values based on a normal distribution. Significance was determined using a Fisher's exact test. Raw and false discovery rate corrected p-values are reported with p-values <0.03 considered significant. (**A**) Contingency table values for total dataset and (**B**) values unique to recent isolates.
(XLSX)

**S3 Table. Total number and enrichment or depletion of nSNPs detected cross all genes.** All 1,495 annotated genes in the J99 reference (AE001439) are reported with the number of within-host nSNPs detected (n = 536 across all genes). Relative z-scores displayed in the table were calculated from weighted values based on gene length. Genes with predicted importation events resulting from intergenic recombination are annotated with an asterisk (*).
(XLSX)

**S4 Table. Importation events across all isolates.** For each of the unique importation events predicted in ClonalFrameML, the table indicates the beginning and ending positions of the region impacted and the gene annotation and ID according to the reference (J99, AE001439). The presence or absence of each importation event across individual isolates is designated as present (1) or absent (0). Importation events that re-occur across multiple lineages are reported in separate lines of the table.
(XLSX)

**S5 Table. Individual SNPs and indels associated with antrum or corpus were not detected in this individual.** Nucleotide positions of SNP or indel unique to recent isolates are reported with contingency tables showing number of isolates from corpus and antrum with and without

a polymorphism in that position as compared to isolates outside that region. Statistical significance was determined with a Fisher's exact test. Raw and false discovery rate corrected p-values are reported with p-values <0.05 considered statistically significant. (**A**)Table of recent SNPs (n = 1,379) unique to corpus isolates (corpus, n = 12) compared to isolates originating from other biopsy sites (other, n = 15). (**B**) Table of recent SNPs (n = 1,379) unique to antrum isolates (antrum, n = 12) compared to isolates originating from other biopsy sites (other, n = 15). (**C**) Table of recent indels (n = 372) unique to corpus isolates (corpus, n = 12) compared to isolates originating from other biopsy sites (other, n = 15). (**D**) Table of recent indels (n = 372) unique to antrum isolates (antrum, n = 12) compared to isolates originating from other biopsy sites (other, n = 15).
(XLSX)

**S6 Table. Short read sequencing metrics and BioSample IDs for the NCBI Sequence Read Archive (SRA).** The reference sequence, sequence platform, NCBI BioProject and BioSample IDs for each of the 43 genomes sequenced in this collection. Average read depth coverage for each genome is reported with isolates below <30 highlighted in red. These isolates were not submitted to the NCBI SRA or included in the analysis.
(XLSX)

**S7 Table. Pairwise comparison data used in Figs 2, 4, 5 and 7.** Genetic distance, shared sites, π values, and time between isolation for each unique pairwise comparison of isolates reported in this study.
(XLSX)

**S8 Table. Primers used in this study.** List of primer sequences used in this study for sequencing and strain construction.
(XLSX)

**S9 Table. Strains used in this study.**
(XLSX)

## Acknowledgments

The authors would like to acknowledge Rick Peek Jr. lab members for collecting, processing, and sharing the samples and isolates used in this study, Katherine Xue for assistance with Breseq and Samtools variant calling, Jesse Domingo and Sherwin Shabdar for their contributions to the CellTool collection of 2D images.

## Author Contributions

**Conceptualization:** Laura K. Jackson, Richard M. Peek, Jr, Nina R. Salama.

**Data curation:** Laura K. Jackson.

**Formal analysis:** Laura K. Jackson, Barney Potter, Sean Schneider, Matthew Fitzgibbon, Kris Blair, Hajirah Farah.

**Funding acquisition:** Nina R. Salama.

**Investigation:** Laura K. Jackson, Barney Potter, Sean Schneider, Kris Blair, Hajirah Farah.

**Methodology:** Laura K. Jackson, Trevor Bedford, Nina R. Salama.

**Project administration:** Nina R. Salama.

**Resources:** Uma Krishna, Richard M. Peek, Jr.

**Software:** Trevor Bedford.

**Supervision:** Nina R. Salama.

**Visualization:** Laura K. Jackson, Barney Potter.

**Writing – original draft:** Laura K. Jackson.

**Writing – review & editing:** Nina R. Salama.

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
