## [Decision Letter · Decision Letter 0]

20 Jul 2020

Dear Nina,

Thank you very much for submitting your manuscript "Helicobacter pylori diversification during chronic infection within a single host generates sub-populations with distinct phenotypes" for consideration at PLOS Pathogens. As with all papers reviewed by the journal, your manuscript was reviewed by members of the editorial board and by several independent reviewers. In light of the reviews (below this email), we would like to invite the resubmission of a significantly-revised version that takes into account the reviewers' comments.

Overall, the referees agreed that the strengths of the manuscript include the biomedical importance of understanding "within infection" factors that contribute to Hp genomic changes that drive Hp-dependent emergence of gastric disease in humans. The premise of the study is well founded, and takes advantage of an important resource (clinical samples) collected from different gastric sites of an individual at two different time points separated by approximately 6 years. There is also documented medical information that documents profound changes in the health status of the individual from whom the strains were collected.

However, an array of serious concerns were expressed by two of the three referees. Referee 2 highlighted several major weaknesses included the inability to identify whether the observed phenotypes reported were due to new, clade- or isolate-specific mutations, as well as the inability to differentiate whether the phenotypes could be ascribed to de novo mutation or recombination. There were also queries as to whether repeated phenotypes were examples of genetic parallelism? Both reviewers 2 and 3 were uneasy about the methods of comparative genome analysis. In addition, major concerns were raised about the reference J99 sequence, which contained many changes in the genome relative to the Genbank sequence of J99, many of which were clustered to changes documented in the clinical isolates reported in this study. Concern was expressed that these changes could substantially inflate the number of unique SNPs observed within the clinical isolates, and thereby potentially confound the author’s interpretations. In addition to these major shortcomings, a variety of additional major and minor items were identified, and are highlighted in the referee comments, that reduced overall enthusiasm for the manuscript.

Overall, although there was enthusiasm expressed by all three referees for the premise and potential importance of the studies reported in this manuscript, there were serious shortcomings that the referees highlighted, that would ultimately require highly extensive additional work to suitably address major concerns.

We cannot make any decision about publication until we have seen the revised manuscript and your response to the reviewers' comments. Your revised manuscript is also likely to be sent to reviewers for further evaluation.

Sincerely,

Steven R. Blanke

Associate Editor

PLOS Pathogens

Alan Hauser

Section Editor

PLOS Pathogens

Kasturi Haldar

Editor-in-Chief

PLOS Pathogens

orcid.org/0000-0001-5065-158X

Michael Malim

Editor-in-Chief

PLOS Pathogens

orcid.org/0000-0002-7699-2064

Overall, the referees agreed that the strengths of the manuscript include the biomedical importance of understanding "within infection" factors that contribute to Hp genomic changes that drive Hp-dependent emergence of gastric disease in humans. The premise of the study is well founded, and takes advantage of an important resource (clinical samples) collected from different gastric sites of an individual at two different time points separated by approximately 6 years. There is also documented medical information that documents profound changes in the health status of the individual from whom the strains were collected.

However, an array of serious concerns were expressed by two of the three referees. Referee 2 highlighted several major weaknesses included the inability to identify whether the observed phenotypes reported were due to new, clade- or isolate-specific mutations, as well as the inability to differentiate whether the phenotypes could be ascribed to de novo mutation or recombination. There were also queries as to whether repeated phenotypes were examples of genetic parallelism? Both reviewers 2 and 3 were uneasy about the methods of comparative genome analysis. In addition, major concerns were raised about the reference J99 sequence, which contained many changes in the genome relative to the Genbank sequence of J99, many of which were clustered to changes documented in the clinical isolates reported in this study. Concern was expressed that these changes could substantially inflate the number of unique SNPs observed within the clinical isolates, and thereby potentially confound the author’s interpretations. In addition to these major shortcomings, a variety of additional major and minor items were identified, and are highlighted in the referee comments, that reduced overall enthusiasm for the manuscript.

Overall, although there was enthusiasm expressed by all three referees for the premise and potential importance of the studies reported in this manuscript, there were serious shortcomings that the referees highlighted, that would ultimately require highly extensive additional work to suitably address major concerns.

Reviewer's Responses to Questions

**Part I - Summary**

Reviewer #1: Long-term infections with the bacterial type-I carcinogen Helicobacter (Hp) have been associated with a broad range of gastric disorders, including gastritis, ulceration, gastric cancer or MALT lymphoma. The manifestation of clinical diseases associated with Hp infection is driven by strain properties and host susceptibility. Previous studies have documented frequent and extensive within-host bacterial genetic variation. To define how within host diversity contributes to phenotypes related to Hp pathogenesis, the authors collected 39 clinical isolates, acquired prospectively from a single subject at two time points and from multiple gastric sites. During the six years separating collection of these isolates, this individual, initially harbouring a duodenal ulcer, progressed to gastric atrophy and concomitant loss of acid secretion. The authors performed whole genome sequence analysis and identified 2,232 unique single nucleotide polymorphisms (SNPs) across isolates and a nucleotide substitution rate of 1.3x10 -4 substitutions/site/year. They also analysed phenotypic differences in bacterial morphology, ability to induce inflammatory cytokines, and mouse colonization. For example, higher inflammatory cytokine induction were correlated with polymorphisms in the Cag PAI protein CagY, while rod morphology in a subgroup of recent isolates mapped to eight mutations in three distinct helical cell shape determining (csd) genes. The presence of subgroups with unique genetic and phenotypic properties suggest complex selective forces and multiple sub-niches within the stomach during chronic infection. Together, this is an interesting paper, which will raise considerable attention in the community.

Reviewer #2: This study takes advantage of a remarkable collection of H pylori isolates from a chronic infection lasting six years. Given what’s generally known about the standing diversity, mutation rates, and evolutionary rates of H. pylori, the fact that the authors observe population divergence following WGS is not surprising, but it's certainly worth documenting carefully. The discovery that late isolates tend to stimulate higher IL-8 secretion and have altered morphologies that may be adaptive is interesting and sets the stage for further study about the mechanisms of host adaptation, ideally with isolates differentiated by fewer genetic differences.

There are two significant limiters to this study. The first is that all conclusions are drawn from two sampling periods with no intermediates, and the manuscript relies upon, and often overstates, correlations drawn from only these two periods. This must be acknowledged and reconciled. The second is that many conclusions are drawn on the basis of the amount of pairwise genetic differences between strains, and this requires both greater rigor and appropriate phylogenetic context. I make recommendations about improving the rigor of variant calling follow; however, regardless if these methods don’t change the among-isolate variance much, the entire study must be grounded in phylogeny rather than sets of pairwise differences that largely reflect time and effects of a few outliers (eg strain/clade A2). That is, it’s fine to say that variation among isolates increases over time, and interesting that this variation does not correlate with point of isolation, but this variation likely evolved from a single clone and hence most mutations differentiating strains are non-independent due to shared ancestry.

The key question that’s not clearly resolved is what are the new, clade- or isolate-specific mutations that might account for the interesting phenotypes you observe? Further, are these caused by de novo mutation or recombination? Are these repeated phenotypes also potentially examples of genetic parallelism? Suggestions are offered but are not as convincing as they could be with more rigorous phylogenomic inference of driver mutations.

Reviewer #3: This paper by the group of Nina Salama describes the genome-wide characterization of sequence variation in a collection of H. pylori isolates taken from one individual at two different time points. The collection of strains is unique due to the extensive sampling at two time points with a six year interval. It is also of particular interest, because one of the strains isolated from this patient at the first time point is H. pylori strain J99, one of the best characterized H. pylori strains. Subgenomic (e.g. microarray) analyses of a selection of these isolates were published before (Israel, Salama et al., PNAS 2001). To apply whole genome sequencing to these isolates is straightforward and the authors are to be commended for taking on the endeavor of reanalyzing this important set of strains, using Illumina MiSeq technology. The results of the genome comparisons were used as a basis for designing additional experiments which a focus on phenotypical traits successfully studied in the Salama laboratory, such as bacterial cell shape, IL-8 induction, and virulence in a mouse model.

**Part II – Major Issues: Key Experiments Required for Acceptance**

Reviewer #1: no

Reviewer #2: Major

Overall the introduction describes an array of mechanisms of H pylori variation but not a sense of what is known about the typical diversity within or among infections. Please provide this background and relate your results to what’s been shown.

* Methods and rigor of sequencing (what was average coverage, estimated genome size per isolate?) and variant calling require clarification.

While breseq is a standard-bearer, the key question is how many sites are informative – since some are prone to variation regardless of variant calling method even for extremely well studied reference genomes -- and how many of those remaining can be used to rigorously differentiate isolates. Sampling bias is a problem for some analyses because the 1B clade is n=1. The IL-8 phenotype and cell shape phenotypes is interesting but complicated.

Method for distinguishing between ancestral and recent diversity (Table 1) is not described and needs at the minimum a phylogenomic analysis if not ancestral reconstruction. Recent diversity could actually have existed earlier without detection, right?

- Repeated, different (I think they are unique, but clarify?) mutations in the same gene (Table 2) could result from diversifying selection in that trait but also could result from poor assembly or read-mapping. Recombination in these regions could also confound inference. Please comment on these possibilities.

Fig 2 – how are SVs accounted in your estimate of pairwise nucleotide variation? Looks like difference is driven by enrichment of a more divergent set of isolates from the first sampling point.

Fig 5: panel C is disingenuous. There’s no relationship between numbers of SNPs and IL8 induction, this is very clearly a false correlation where time and likely a few mutations are the causative variable.

Reviewer #3: Unfortunately, the comparative genome analysis is not (yet) State of the Art and so the study currently fails to deliver on its promise and potential. The most important concerns are:

1.) L269//506-509: The authors chose to use an “agnostic” (their word) approach, i.e. not to distinguish between mutation and polymorphisms introduced by recombination. In my view, this is a very bad choice, since it makes the results (e.g. the frequency of polymorphisms) impossible to compare with other studies, and likewise very hard to interpret. A quick analysis of the sequences shows ample evidence of recombination in the later set of strains. Methods for accurate phylogenetic inference of recombination in bacterial genomes, such as Gubbins or ClonalFrameML, are now available and widely used. Why choose a dendrogram over a phylogenetic tree that would have provided a more accurate representation of the evolutionary relationships between the isolates? Moreover, a brief inspection the VCF file provided on the Github repository indicates that many SNPs present in the recent group of isolates are tightly clustered together, suggesting that a majority of the diversity could actually come from recombination. In any case, more technical information should be provided concerning the construction of the dendrogram (i.e. is it just hierarchical clustering of pairwise SNP count/genetic distance or was a substitution model used?).

2.) L189/Table 1/Table S1A. It appears that strain J99 was resequenced along with the other isolates. Suprisingly, the new J99 sequence displays hundreds of differences compared to the Genbank sequence of J99. These differences must either be sequencing errors in the reference sequence or in the Illumina data, or the sequences do not come from the same strain. Furthermore, and of even more concern, most the SNPs observed between the “old” and “new” versions of J99 also appear in all the isolates in this study. This raises substantial doubts about all quantitative statements regarding sequence changes during the six years of infection. There are several possible explanations for this, one possibility is that the clone sequenced by Richard Alm and colleagues was substantially different from the current J99. Since the patient apparently did have a mixed infection (since there is so much recombination), this is possible. Alternatively, the old sequence may contain many errors, but then the point of reference for all isolates should be the “new” J99 sequence, not the old one. This issue appears to inflate the number of unique SNPs observed by around 20%, which is very significant. It would probably be best to complete the new genome sequence of J99 (e.g. by adding PacBio or Oxford Nanopore long reads) and to use this as a reference for the analysis. This is a major issue affecting the validity of all comparisons and must be addressed.

3.) L191: Both mutation and recombination are considered here, thus using “polymorphism” instead of “mutation” would be more accurate.

4.) L227-228: How different is the pool of within-host variable OMPs between the old and recent group of isolates?

5.) L258-259: Are these four isolates all from the recent group? The 23S rRNA is typically fairly conserved. How do you explain that polymorphisms related to antibiotic resistance appeared at a significant frequency in the population with no known history of antibiotic treatment?

6.) Figure 4/5/6/7: The unit used on the x-axis may not be correct. On the NextStrain platform, the scale indicates “Divergence”, which seems to correspond to the accumulation of changes since the common ancestor and not a rate like “substitution per site per year” seems to indicate.

7.) Figure 4/5/6/7: Isolates from the duodenum, corpus and antrum regions appear in both subgroups 2A and 2B. This should be discussed.

8.) Figure 4/5/6/7: The common ancestor of the recent group of isolates appears to only date back to a few months before the sampling date (using the year scale on the Nextstrain platform). How is it explained if the population has been evolving from the population of 6 years ago without a major bottleneck?

**Part III – Minor Issues: Editorial and Data Presentation Modifications**

Reviewer #1: Minor points for improvement:

1.) Table 2: How many SNPs were seen per individual Hp gene?

2.) Just one SNP per Hp strain and gene or multiple SNPs in any particular gene, e.g. hopQ?

3.) If multiple SNPs, could they be the result of intrachromosomal recombination?

4.) In Fig. 5 you mention a > 20% increase, but in Fig. 9 you refer to >20% decrease. Please clarify.

5.) How do you compare the mutation rates to that in previously published papers from isolates of chronically infected patients?

Reviewer #2: Minor

66 “sub niches” awkward and not a standard term.

77 please provide specific details about what’s known about the extent of diversification of H pylori during a typical infection. Important for setting baseline

81-83 same comment about cagAY allelic variation – how much?

129 what’s the quality of J1 genome? How was it produced? You’ve now resequenced it effectively – how many calls are shared, derived and thus actually in the J1 isolate?

176 92% ANI is beyond the level of species distinction, even genus – this seems inconceivable. Sequencing and mapping details need to be provided up front to justify this claim

184-188 writing is awkward

223: “functional classes assigned by the microbial genome database (MGDB) add reference.” What do you mean? Definitely add the reference!

230 you state others have found variation in OMPs but don’t provide references. Be more specific about the kind and consequences of this variation.

Fig 3 statistical enrichment of functional categories should be conservative because of numerous potential biases; omit those at p=0.03.

263 “but no mutations indicative of additional antibiotic resistance were discovered.” Vague. Describe methods esp in light of the broad challenge to identify AMR determinants from genomes.

Reviewer #3: L676: It iss not quite clear how experimental variation was assessed. Were multiple tests performed for every individual isolate? What are “pooled replicates from two independent experiments”?

Fig. 1: The information contained in Fig. 1 is also contained in the text, consider deleting it to conserve space.

The reference section needs careful checking for completeness and formatting of citations.

PLOS authors have the option to publish the peer review history of their article (what does this mean?). If published, this will include your full peer review and any attached files.

Reviewer #1: No

Reviewer #2: No

Reviewer #3: No
---

## [Decision Letter · Decision Letter 1]

22 Oct 2020

Dear Nina,

We are pleased to inform you that your manuscript 'Helicobacter pylori diversification during chronic infection within a single host generates sub-populations with distinct phenotypes' has been provisionally accepted for publication in PLOS Pathogens. The referees were highly impressed with the manner in which you addressed the comments and concerns expressed in regard to the initial submission, and believed that the findings reported in the revised manuscript represents an important advance in the field of Helicobacter pylori infection biology.

Best regards,

Steven R. Blanke

Associate Editor

PLOS Pathogens

Alan Hauser

Section Editor

PLOS Pathogens

Kasturi Haldar

Editor-in-Chief

PLOS Pathogens

orcid.org/0000-0001-5065-158X

Michael Malim

Editor-in-Chief

PLOS Pathogens

orcid.org/0000-0002-7699-2064

Reviewer Comments (if any, and for reference):

Reviewer's Responses to Questions

**Part I - Summary**

Reviewer #1: accepted

Reviewer #2: I was impressed by the unique dataset, detailed genomic study, and careful follow up analyses of genetic variation evolving during prolonged H pylori infection. With this highly responsive revision that tackles each comment from our reviews, I'm now even more so. I think this will be an important contribution to our understanding of bacterial evolution during chronic infections in general, as well as the unique dynamics of H pylori infections.

Reviewer #3: The authors have fully addressed all points raised in my review.

**Part II – Major Issues: Key Experiments Required for Acceptance**

Reviewer #1: accepted

Reviewer #2: (No Response)

Reviewer #3: (No Response)

**Part III – Minor Issues: Editorial and Data Presentation Modifications**

Reviewer #1: accepted

Reviewer #2: unnecessary comma in line 362.

Reviewer #3: (No Response)

PLOS authors have the option to publish the peer review history of their article (what does this mean?). If published, this will include your full peer review and any attached files.

Reviewer #1: No

Reviewer #2: No

Reviewer #3: No

---

## [Editor Report · Acceptance letter]

9 Dec 2020

Dear Dr. Salama,

We are delighted to inform you that your manuscript, "Helicobacter pylori diversification during chronic infection within a single host generates sub-populations with distinct phenotypes," has been formally accepted for publication in PLOS Pathogens.

Best regards,

Kasturi Haldar

Editor-in-Chief

PLOS Pathogens

orcid.org/0000-0001-5065-158X

Michael Malim

Editor-in-Chief

PLOS Pathogens

orcid.org/0000-0002-7699-2064